# RL-ViGen: A Reinforcement Learning Benchmark for Visual Generalization

**Zhecheng Yuan**[1,3,4*], **Sizhe Yang**[2,3*], **Pu Hua**[1,3], **Can Chang**[1,3,4],
**Kaizhe Hu**[1,3,4], **Huazhe Xu**[1,3,4]

[1] Tsinghua University, [2] University of Electronic Science and Technology of China,
[3] Shanghai Qi Zhi Institute, [4] Shanghai AI Lab
yuanzc23@mails.tsinghua.edu.cn, huazhe_xu@mail.tsinghua.edu.cn

## Abstract

Visual Reinforcement Learning (Visual RL), coupled with high-dimensional observations, has consistently confronted the long-standing challenge of out-of-distribution generalization. Despite the focus on algorithms aimed at resolving visual generalization problems, we argue that the devil is in the existing benchmarks as they are restricted to isolated tasks and generalization categories, undermining a comprehensive evaluation of agents' visual generalization capabilities. To bridge this gap, we introduce RL-ViGen: a novel **R**einforcement **L**earning Benchmark for **Vi**sual **Gen**eralization, which contains diverse tasks and a wide spectrum of generalization types, thereby facilitating the derivation of more reliable conclusions. Furthermore, RL-ViGen incorporates the latest generalization visual RL algorithms into a unified framework, under which the experiment results indicate that no single existing algorithm has prevailed universally across tasks. Our aspiration is that RL-ViGen will serve as a catalyst in this area, and lay a foundation for the future creation of universal visual generalization RL agents suitable for real-world scenarios. Access to our code and implemented algorithms is provided at `https://gemcollector.github.io/RL-ViGen/`.

## 1 Introduction

Visual Reinforcement Learning (RL) has attained remarkable success across a plethora of domains [39, 43, 14]. A diverse range of techniques has been implemented to tackle not only the trial-and-error learning process but also the complexity arising from high-dimensional input data. Notwithstanding these successes, a fundamental challenge confronting visual RL agents persists — achieving generalization.

To overcome this obstacle, several visual RL generalization benchmarks have emerged, including Procgen [6], Distracting Control Suite [47], and DMC-GB [21]. While these benchmarks have been indispensable to visual RL generalization progress, they are not exempt from inherent limitations that pose challenges to further development. Procgen offers a diverse distribution of environment configurations and visual appearances. However, it is limited to video games with non-realistic images and low-dimensional discrete action spaces, resulting in a significant gap between its environments and real-world scenarios. Another instance, DMC-GB, is sometimes treated as a golden standard for many state-of-the-art visual generalization algorithms.

---

*equal contribution

37th Conference on Neural Information Processing Systems (NeurIPS 2023) Track on Datasets and Benchmarks.

|  | **Training** | **Generalized Visual Scenarios** |
|---|---|---|

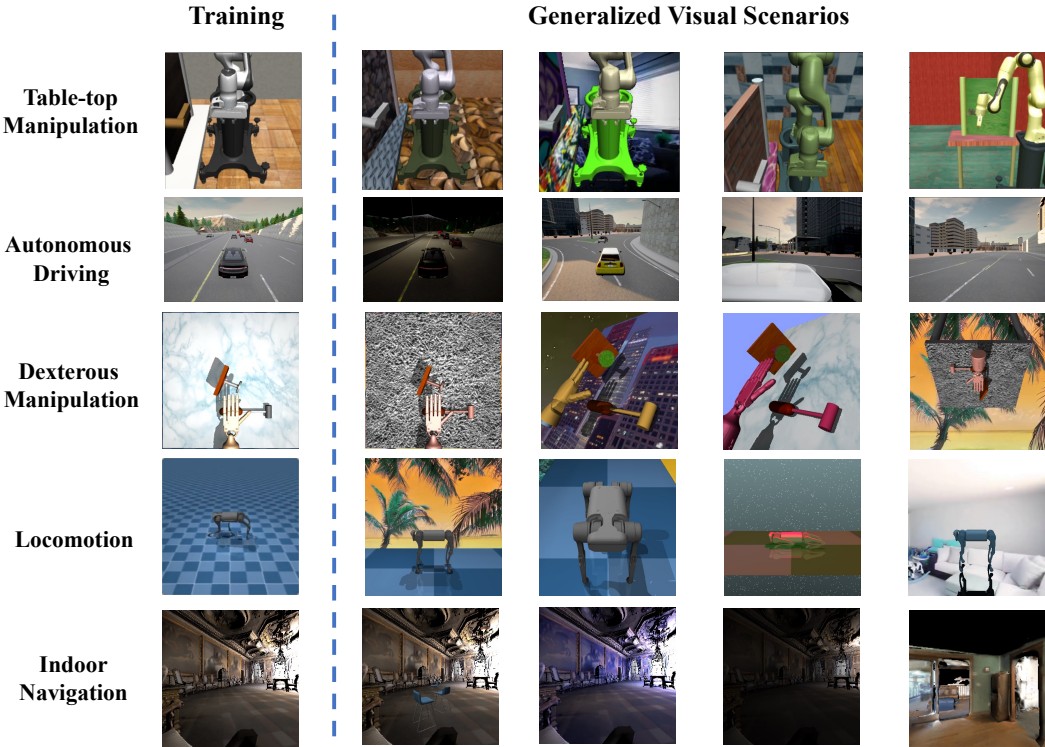

Figure 1: **The novel RL benchmark for visual generalization.** We show that RL-ViGen supports a wide range of tasks with different generalization categories. The algorithms can be evaluated more comprehensively and achieve more convincing experimental results.

Nevertheless, the narrow scope of task classes and generalization categories in existing setups cannot thoroughly and comprehensively evaluate the agent's generalization ability. In addition, although Distracting Control Suite contains two generalization types, it falls short in diversity and complexity. The prevailing trend in this field is to showcase the superiority of proposed algorithms on these benchmarks, which adversely poses a certain risk of promoting overfitting to these benchmarks, rather than discovering algorithms potentially beneficial for solving real-world problems.

In this paper, we introduce a novel **R**einforcement **L**earning benchmark for **Vi**sual **Gen**eralization (RL-ViGen), presenting numerous merits over existing counterparts. Our benchmark integrates a spectrum of task categories with realistic image inputs, including table-top manipulation, locomotion, autonomous driving, indoor navigation, and dexterous hand manipulation, allowing for a more comprehensive evaluation of the agents' efficacy. Moreover, by incorporating various key aspects in visual RL generalization, such as visual appearances, lighting changes, camera views, scene structures, and cross embodiments, RL-ViGen enables a comprehensive examination of agents' generalization ability against distinct visual conditions.

It is noteworthy that we provide a unified framework that encompasses various state-of-the-art visual RL and generalization algorithms with the same optimization scheme for each approach. The framework not only promotes fair benchmarking comparisons but also lowers the entry barrier for devising novel approaches.

In summary, our contributions are as follows: **1)** we propose a novel visual RL generalization benchmark RL-ViGen with diverse, realistic rendering tasks and numerous generalization types; **2)** we implement and evaluate various algorithms within a unified framework, enabling a comprehensive analysis of their generalization performance; **3)** we conduct comprehensive and extensive experiments to demonstrate the distinct performance of existing approaches when tackling diverse tasks and generalization types, and highlight the benefits and the limitations of current generalizable visual RL algorithms. With all the contributions combined, RL-ViGen may pave the way for further advancements in visual RL generalization, ultimately leading to more robust and adaptable algorithms for real-world applications.

Table 1: **Generalization Categories.** The following table outlines the types of generalization incorporated within each task. Except for categories considered as not applicable (N/A) (e.g., for locomotion, changes in scene structures are not required), all potential types are included.

| Generalization, Categories | Visual Appearances | Camera Views | Lighting Changes | Scene Structures | Cross Embodiments |
|---|---|---|---|---|---|
| **Autonomous Driving** | ✓ | ✓ | ✓ | ✓ | ✓ |
| **Table-top Manipulation** | ✓ | ✓ | ✓ | N/A | ✓ |
| **Indoor Navigation** | ✓ | ✓ | ✓ | ✓ | N/A |
| **Dexterous Manipulation** | ✓ | ✓ | ✓ | N/A | ✓ |
| **Locomotion** | ✓ | ✓ | ✓ | N/A | ✓ |

## 2 RL-ViGen

RL-ViGen consists of 5 distinct task categories, spanning the domain of locomotion, table-top manipulation, autonomous driving, indoor navigation, and dexterous hand manipulation. In contrast to prior benchmarks, RL-ViGen employs a diverse array of task classes for evaluating the agent's generalization performance. We believe that only through comprehensive examination from multiple perspectives can we obtain convincing results. Furthermore, as shown in Table 1, our benchmark offers a wide range of generalization categories, including visual appearances, camera views, variations in lighting conditions, scene structures, and cross embodiments settings, thereby providing a thorough evaluation of algorithms' robustness and generalization abilities.

### 2.1 Environments

**Dexterous manipulation:** Adroit [44] is a sophisticated environment that is explicitly tailored for dexterous hand manipulation tasks. It demands considerable exploration and fine-grained feature capturing due to the sparse reward nature of the environment and the complexity of high-dimensional action space. In RL-ViGen, we have enriched the Adroit environment by integrating diverse visual appearances, camera perspectives, hand types, lighting changes, and object shapes.

**Autonomous driving:** CARLA [9] serves as a realistic and high-fidelity simulator for autonomous driving, which investigates the control capabilities of agents under dynamic conditions. It has been successfully deployed on visual RL settings in prior studies. Contrary to previous work [24], RL-ViGen provides an enhanced range of dynamic weather and more complex road conditions in different scene structures. Furthermore, flexible camera angle adjustments are also included within RL-ViGen.

**Indoor navigation:** As an efficient and photorealistic 3D simulator, Habitat [46] combines numerous visual navigation tasks. Succeeding in these tasks requires the agents to own the capability of scene understanding. RL-ViGen builds upon the *skokloster-castle* scene and proposes additional scenarios with different visual and lighting settings. In addition, the camera view and scene structure are designed to be adjustable.

**Table-top manipulation:** Robosuite [64] is a modular simulation platform designed to support robot learning. It inherently contains interfaces designed to adjust various scene parameters. Recent work [12] has leveraged this platform to test the agent's generalization ability of visual background changes. RL-ViGen further incorporates dynamic backgrounds, adaptive lighting conditions, and options for embodiment variation, refining the simulation to be closer to the real world.

**Locomotion:** DeepMind Control is a popular continuous visual RL benchmark. DMC-GB [22] is developed on it and has become a widely used benchmark for evaluating generalization algorithms. Building upon DMC-GB, RL-ViGen introduces objects and corresponding tasks from sophisticated real-world locomotion and manipulation applications, such as the Unitree, Anymal quadrupedal robots, and the Franka Arm. What's more, RL-ViGen also offers a variety of generalization categories to further enrich this environment.

More detailed implementations and modifications can be found in Appendix B and our codebase.

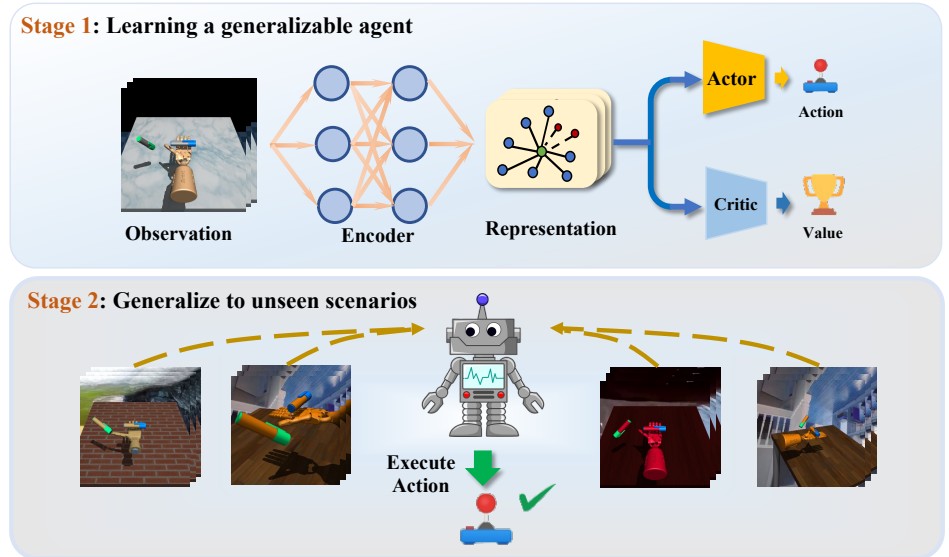

Figure 2: **Generalization procedure.** The agent is first trained in Stage 1 with a certain fixed scenario. Subsequently, in Stage 2, the agent is tested across various visual generalization scenes in a zero-shot manner. The better the agent performs in various scenes of Stage 2, the stronger generalization ability it demonstrates.

## 2.2 Generalization Categories

Here, we emphasize the primary generalization categories utilized in RL-ViGen:

**Visual appearances:** Maintaining effective performance in the presence of altered visual features of objects, scenes, or environments is of vital importance, particularly for visual reinforcement learning. In our benchmark, different components within the environment can be modified with a wide range of colors. Meanwhile, the dynamic video background is also introduced as a challenging setting.

**Camera views:** In the real world, the agents have to cope with camera configurations, angles, or positions that may not align with those experienced during training. We offer access to set the cameras at different angles, distances, and FOVs. In addition, the number of cameras can be adjusted accordingly.

**Lighting conditions:** The change in lighting conditions will occur inevitably in the real world. To equip agents with the ability to adapt to such variations, our benchmark supplies interfaces related to the lighting, such as varied light intensity, colors, and dynamical shadow changes.

**Scene structures:** Mastering the ability of understanding and adapting to different spatial arrangements and organization patterns within various scenes is crucial for a truly generalizable agent. To this end, our benchmark enables modifications in scene structure via adjusting maps, patterns, or introducing extra objects.

**Cross embodiments:** Adapting learned skills and knowledge to different physical morphologies or embodiments is essential for an agent to perform well across various platforms or robots with different kinematic structures and sensor configurations. Therefore, our benchmark also provides access to modify the embodiment of trained agents in the aspects of model types, sizes, and other physical properties.

## 3 Algorithmic Baselines for Generalization in Visual RL

### 3.1 A Unified Framework

Another key contribution of our work is the implementation of a unified codebase to support comparison among various visual RL algorithms. In previous studies, different algorithms adopt distinct optimization schemes, RL baselines, and hyperparameters. For example, SRM [25] and SVEA [23] rely on SAC-based RL algorithms, while PIE-G [61] utilizes a DDPG-based approach. Moreover, mi-

nor different implementations could substantially impact the final performance. Therefore, providing a unified framework is of great importance in this domain, enabling more persuasive conclusions to be drawn from evaluating algorithms across a consistent framework and diverse tasks.

## 3.2 Visual RL Algorithms

In our benchmark, we assemble eight leading visual RL algorithms and apply the same unified training and evaluation framework. **DrQ-v2** [57] is the prior state-of-the-art DDPG-based model-free visual RL algorithm in terms of sample efficiency. **DrQ** [32] is another SAC-based sample efficient visual RL algorithm, which is the base of DrQ-v2. **CURL** [33] utilizes a SimCLR-style [5] contrastive loss to obtain better visual representations. **VRL3** [50] is the state-of-the-art algorithm in Adroit tasks with human demonstrations. The other four algorithms concentrate on achieving robust representations. **SVEA** [23] employs the Q-value of un-augmented images as the target objective while utilizing data augmentation for reducing the Q-variance; **SRM** [25] adopts augmentation in the frequency domain to selectively eliminate a part of the observation frequency; **PIE-G** [61] incorporates ImageNet [8] pre-trained model to further boost the generalization ability; **SGQN** [3] identifies critical pixels for decision-making via integrating with the saliency map.

## 4 Experiments

In this section, we try to investigate the generalization ability of different approaches in the proposed RL-ViGen benchmark. As shown in Figure 2, all agents are trained in the same fixed training environment and evaluated within various unseen scenarios in a zero-shot manner. The training sample efficiency and asymptotic performance are shown in Appendix F.4. For each task, we evaluate over 5 random seeds and report the mean scores and 95% confidence intervals. In terms of each trained environment, we present the aggregated scores of the multiple subtasks. The detailed and extensive experimental results can be found in Appendix B and F. The visualization of each environment and generalization types are shown in Appendix D.1.

### 4.1 Visual Appearances and Lighting Changes

#### 4.1.1 Indoor Navigation

Within the Habitat platform, we choose the *ImageNav* task and modify the 3D scanned models to introduce novel scenarios with various visual appearances and lighting conditions. We conduct 10 evaluations in each of the 10 selected scenarios (100 trials in total). In contrast to most existing benchmarks, the Habitat-rendered images are captured from a first-person viewpoint by the high-performance 3D simulator. Hence, it can deliver a visualization more akin to real-world scenes. As shown in Figure 3, the superior performance of PIE-G can be attributed to the integration with the ImageNet pre-trained model, equipping PIE-G with a wealth of authentic images and enabling it to handle these scenarios

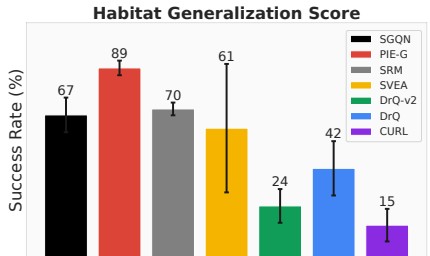

Figure 3: **Generalization score of indoor navigation.** We present the success rate of each method. The result indicates that PIE-G achieves better generalization performance on Habitat.

more efficiently. Conversely, consistent with the conclusion drawn from Section 4.1.2, SGQN, which intends to segment the centered agent via eliminating the redundant background, is proved ineffective in these object-rich and first-person view tasks.

#### 4.1.2 Autonomous Driving

Regarding CARLA, we adopt the reward function setting in Zhang et al. [62] and apply a first-person perspective to better resemble real-world driving conditions. As shown in Figure 11 in Appendix D.1, this environment is divided into three levels: *Easy*, *Medium*, and *Hard*. The main modifications involve varying factors such as rainfall intensity, road wetness, and lighting. The higher the disparity from the training scenarios, the more challenging the difficulty level. In this task, one of its distinctive

features is that the input image frequency undergoes considerable changes. Consequently, the SRM approach, which applies data augmentation in the frequency domain, demonstrates the best performance as it can adapt to the input images with varying frequencies. While PIE-G incorporates the ImageNet pre-trained model, its source training images mainly possess higher-frequency features, thus suffering from suboptimal generalization when facing low-frequency scenarios (e.g., dark night). Moreover, SGQN, which extracts salient information, exhibits a decrease in performance when faced with visually rich scenes where the controlled agent is not present in the observed frame. It also should be noted that DrQ gains a degree of generalization ability in this environment. Our observations suggest that since DrQ is a SAC-based algorithm, it tends to be prone to entropy collapse [57]. This implies that the trained agent only produces a single distribution of action in response to diverse image inputs.

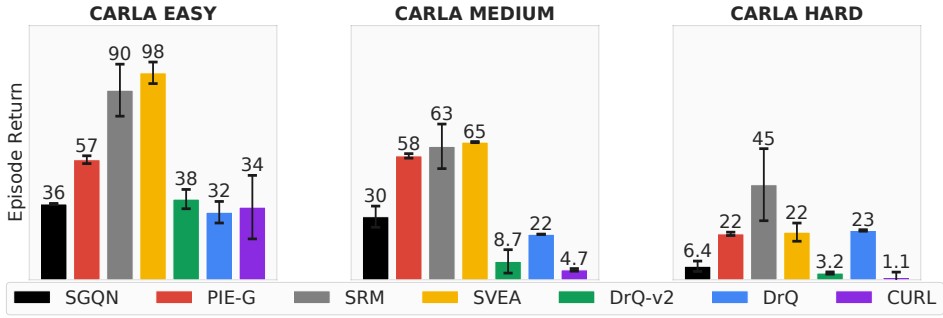

Figure 4: **Aggregated generalization score of autonomous driving.** We present the aggregated return of each method. SRM exhibits better performance to adapt to scenarios where image frequency varies dramatically.

### 4.1.3 Dexterous Hand Manipulation

In the Adroit environment, we assess the performance of each approach in three single-view tasks: *Door*, *Hammer*, and *Pen*. Since DrQ-v2 and DrQ barely perform well in these challenging environments, we utilize VRL3 [50], the state-of-the-art method in this domain, as the base algorithm and the visual RL approaches in RL-ViGen are re-implemented upon it.

With respect to sample efficiency, it is commonly believed that applying strong augmentation will negatively affect sample efficiency. However, as illustrated in Figure 5 in Appendix F.4, it is worth noting that since VRL3 specifically designs a safe Q mechanism to prevent potential Q divergence for this environment, the generalization algorithms applying strong augmentation can achieve performance comparable to those using only random shift.

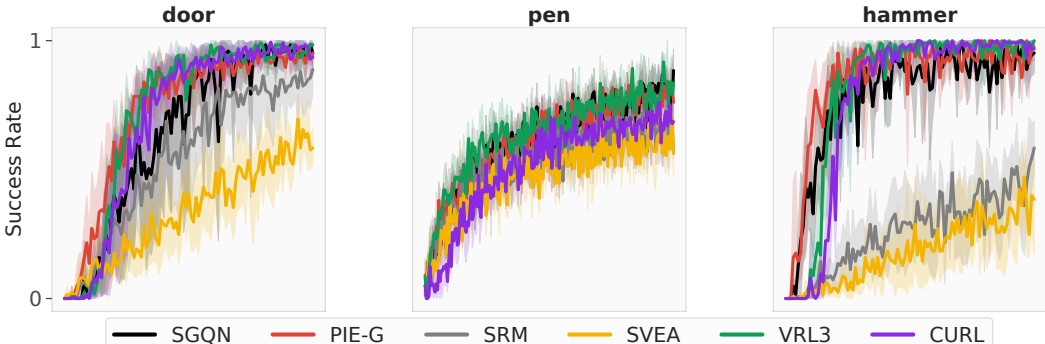

Figure 5: **Sample efficiency of Adroit.** The success rate of each algorithm. We normalize the training steps into (0, 1). The approaches with strong augmentation can also gain comparable performance.

As for generalization, Adroit tasks require agents to identify fine-grained features for dexterous and sophisticated manipulation. Therefore, PIE-G, which leverages ImageNet pre-trained models

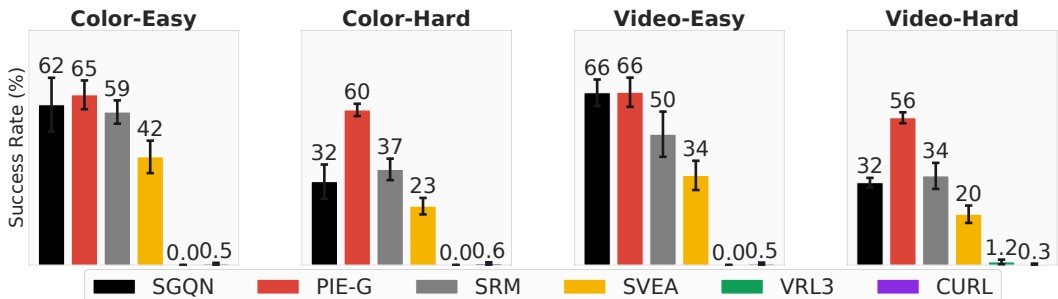

Figure 6: **The aggregated generalization score of dexterous manipulation.** We present the aggregated success rate of each method. PIE-G equipped with the ImageNet pre-trained model exhibits better adaptability to Adroit tasks which necessitate fine-grained information capture.

to capture detailed information, demonstrates the effectiveness of assisting the learned agent in executing downstream tasks, particularly in the *hard* setting. Moreover, as illustrated in Figure 6, the absence of additional objectives to mitigate the effect of visual changes causes both VRL3 and CURL to struggle in adapting to novel visual situations in these demanding tasks.

## 4.2 Scene Structures

Generalizable agents that are capable of delivering robust performance across diverse scene structures are essential for potential broad real-world applications. We select CARLA as the testbed for evaluating the generalization of scene structures. The agents are trained in standard training scenarios (highways), and tested in more complex structure settings, including narrow roads, tunnels, and roundabouts with *HardRainSunset* weather conditions. As shown in Figure 7, the performance of all algorithms falls short of expectations, suggesting that the current visual RL algorithms and generalization approaches are not adequately robust to scene structural changes. More in-depth investigations must be pursued in order to enhance the generalization ability of trained agents to perceive the changing scene structures.

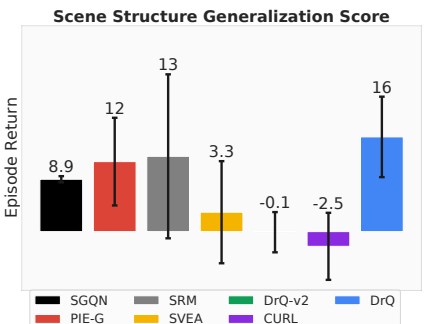

Figure 7: **Generalization score of Scene Structure.** Across this category of generalization, all algorithms demonstrate unsatisfactory performance.

## 4.3 Camera Views

We proceed to evaluate the generalization in terms of camera views in the Adroit Environment. As illustrated in Figure 8, under the *Easy* setting, PIE-G and SGQN exhibit leading generalization capabilities with respect to camera view, while other algorithms also demonstrate some

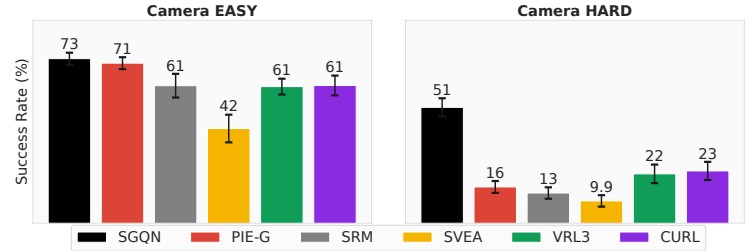

Figure 8: **Generalization score of Camera Views.** SGQN indicates advantageous generalization ability across different levels in camera-view generalization.

degree of generalization due to the use of random shift augmentation. However, in the *Hard* setting, which introduces substantial changes in camera position, orientation, and field of view (FOV), nearly all algorithms, except for SGQN, lose their generalization ability. The exceptional performance of SGQN is mainly due to its heavy reliance on producing saliency maps, which enhances the agent's

self-awareness of object geometry and relative positioning. Hence, this property strengthens its generalization performance even in the face of major camera view alterations.

### 4.4 Cross Embodiments

Addressing the embodiment mismatch from visual input is crucial, as the embodiment composes a substantial portion of the image and significantly influences robot behavior of interacting with the world. To investigate this type of generalization, Robosuite is employed as the evaluation platform. We utilize the OSC_POSE controller [41] during training to facilitate the maintenance of action space dimensions and their respective meanings. Then, the trained agents transfer from Panda Arm to two different morphologies: KUKA IIWA and Kinova3. As illustrated in Figure 9, the overall performance of all algorithms is suboptimal; however, generalization-based methods, which contain more diverse information during training, exhibit a slight advantage over those primarily focused on sample efficiency in the cross-embodiment setting.

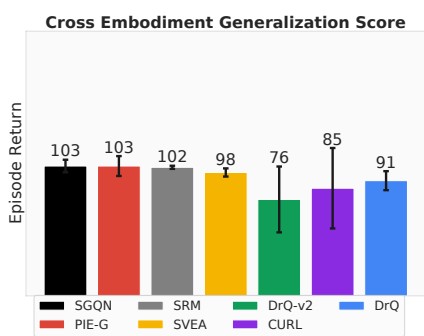

Figure 9: **The aggregated generalization score of Cross Embodiments.** No algorithm has demonstrated the capability to manage the cross-embodiment generalization yet.

## 5 Discussion

In summary, our experiments reveal that the findings based on previous benchmarks may not accurately reflect the actual progress, leading to a distorted perception of the situation; those advanced visual RL algorithms, previously perceived as cutting-edge, display less efficacy within RL-ViGen. We summarize the main takeaways as follows:

**Takeaway 1.** The experimental results reveal the varying generalization performance of different visual RL algorithms in distinct tasks and generalization categories, with no single algorithm demonstrating universally strong generalization abilities.

**Takeaway 2.** Solely enhancing training performance fails to guarantee an improvement in the generalization ability. Although DrQ(v2) and CURL exhibit high sample efficiency during training and even achieve better asymptotic performance (Appendix F.4), their performance in various generalization scenes has yet to reach a satisfactory level. Therefore, when attempting to improve the generalization ability of an agent, it is crucial to introduce additional inductive biases to aid the training process.

**Takeaway 3.** An effective generalizable visual RL agent must demonstrate exceptional performance across multiple generalization categories. Previous work has primarily focused on generalization concerning visual appearances, while our experiments reveal considerable shortcomings of existing algorithms in the setting of cross embodiments and scene structures. These underperforming generalization categories go beyond altering the observation space within the Markov Decision Process (MDP); they also bring modifications to the action space and transition probabilities, thus presenting the agent with extra challenges.

**Takeaway 4.** Each generalization algorithm possesses its own unique strengths. Notably, PIE-G demonstrates superior performance with respect to visual appearances and lighting condition changes, while SRM, under significant image frequency variations, exhibits remarkable robustness. SGQN retains its generalization capacity when facing considerable camera view alterations. In addition, SVEA, without the need for additional parameters and with only minimal modifications, can achieve a certain level of generalization abilities. We hypothesize that stronger performance might be attained through a fusion of different algorithms, such as utilizing pre-trained models with frequency-based augmentation to induce further improvement.

Combined with the takeaways, we hope that an algorithm's success in RL-ViGen can indicate its potential applicability in more complex and unpredictable real-world scenarios. In the future, a

holistic and multi-dimensional approach, encompassing aspects such as scene structures, camera views, and cross embodiments, is critical for fostering truly generalizable agents capable of navigating in varied and dynamic real-world environments. Equally, the design of more sophisticated and realistic training environments which enable to reflect the complexity of real-world conditions, can also serve as a crucial area for future explorations.

# 6 Related Work

**RL benchmarks.** There exists a multitude of mature benchmarks aiming for evaluating reinforcement learning algorithms [4, 51, 38, 10, 17, 40, 55]. For instance, Atari [2] and Gym-MuJoCo [48] are exemplary benchmarks in deep reinforcement learning. In other subdomains, D4RL [15] serves as a popular benchmark for offline RL algorithms, while URLB [34] provides an evaluating platform with respect to unsupervised RL algorithms. MetaWorld [59] is often used to evaluate multi-task and meta-learning scenarios. SafetyGym [45], meanwhile, is predominantly applied for testing Safe RL algorithms. Recently, MineDojo [13] benchmarks embodied agents in exploration and multi-task domains. Contrasting to these benchmarks, RL-ViGen distinguishes itself by incorporating a variety of task classes and an array of generalization categories and primarily focuses on evaluating agents' visual generalization abilities.

**Generalization.** How to endow models' generalization abilities is a pivotal topic in machine learning. In computer vision, well-established benchmarks are available for exploring generalization problems [54, 53, 58, 31]. While several approaches have been proposed in RL and robotics to tackle such issues [23, 61, 25, 3, 63, 29, 30, 28, 56, 60], the benchmarks in use are relatively immature [11, 6, 47, 22], fraught with numerous limitations, and lacking a unified framework for comparison. For example, Procgen [6] is a widely used benchmark for quantifying the agents' generalization abilities. However, Procgen remains a video game platform, with the human-imaged world rather real-world counterparts, offering limited assistance for agents' generalization in real-world scenarios. The Distracting Control Suite [47] and DMC-GB [21], building upon DM-Control [49], introduce some types of visual distractions. Nevertheless, their tasks are solely focused on locomotion, and there remains a substantial gap between this simulated environment and real-world scenarios. By contrast, RL-ViGen encompasses various forms of generalization, exhibits a high degree of photo-realism, and includes a diverse range of tasks. Avalon [1] is another valuable benchmark for RL generalization. It shares a unified world dynamics and task structure, making it highly suitable as a benchmark for in-distribution generalization. Contrary to Avalon, which is mainly concerned with task-level generalization, RL-ViGen mainly focuses on out-of-distribution generalization, with a specific concentration on the visual aspects of generalization.

# 7 Conclusion, Limitations, and Future Work

In this work, we propose a novel Reinforcement Learning benchmark for Visual Generalization (RL-ViGen), a comprehensive benchmark for evaluating the visual generalization abilities of trained agents. RL-ViGen stands apart from existing benchmarks by boasting a broader diversity of tasks and generalization categories, which in turn fosters more persuasive conclusions. According to the quantitative experimental results from RL-ViGen, we note that, as of now, there are no existing generalization algorithms that can adeptly manage all tasks and generalization types. It is our expectation that the advent of RL-ViGen will bring fresh perspectives to the research community, and stimulate the advancement of agents that can truly exhibit overall visual generalization capabilities.

**Limitations.** The agents trained through RL-ViGen have not yet been evaluated in real-world scenarios. In our future work, we would like to build certain real-world tasks to demonstrate the value that RL-ViGen can provide in developing generalizable agents for real-world applications.

# 8 Acknowledgements

This work is supported by research program 2022ZD0161700.

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

# Appendix

## A    Visual Reinforcement Learning Baselines

**DrQ:**    This model-free, off-policy reinforcement learning algorithm, is based on Soft Actor-Critic (SAC) [19]. DrQ enhances training stability via applying data augmentation to regularize the Q value of state-action pairs. The key of DrQ is to promote similarity between augmented state-action pairs. The Q-regularization technique is shown in Eq 1, where $K$ is the number of samples, $\mathcal{T}$ is the collection of augmentation.

$$\mathbb{E}_{\substack{s\sim\mu(\cdot) \\ a\sim\pi(\cdot|s)}} [Q(s,a)] \approx \frac{1}{K}\sum_{k=1}^{K} Q\left(f\left(s^*,\nu_k\right),a_k\right) \text{ where } \nu_k \in \mathcal{T} \text{ and } a_k \sim \pi\left(\cdot \mid f\left(s^*,\nu_k\right)\right) \quad (1)$$

**DrQ-v2:**    An improved version of DrQ. DrQ-v2 fuses essential elements from the DDPG algorithm with data augmentation to strengthen visual RL agents' performance. DrQ-v2 also incorporates techniques such as n-step return and target critic, leading to commendable results in most of the medium and hard level DM-Control tasks. The TD-target is defined as follows, where $x_{t+n}$ is the n-step observation, $\boldsymbol{a}_{t+n}$ is the n-step action, and $\bar{\theta}_{1,2}$ is the Q-target networks:

$$y = \sum_{i=0}^{n-1}\gamma^i r_{t+i} + \gamma^n \min_{k=1,2} Q_{\bar{\theta}_k}\left(aug(x_{t+n}),\boldsymbol{a}_{t+n}\right) \quad (2)$$

**CURL:**    CURL integrates contrastive learning methods into the reinforcement learning training process. The auxiliary contrastive loss (Eq 3) allows the agent to obtain better image representation during training, thus mitigating the optimization difficulty under high-dimensional inputs. In our implementation, we only apply a single encoder to produce visual representations instead of two polyak-averaging encoders. This alteration improves the sample efficiency of CURL and put it on a comparable performance with DrQ-v2. More experiments are shown in Appendix F.3.

$$\mathcal{L}_q = \log \frac{\exp\left(q^T W k_+\right)}{\exp\left(q^T W k_+\right) + \sum_{i=0}^{K-1}\exp\left(q^T W k_i\right)} \quad (3)$$

**PIE-G:**    PIE-G proposes a simple yet effective method, combining ImagNet pre-trained visual representations with the early layer and updates of BatchNorm statistical parameters to further enhance the generalization ability of the agent.

**SVEA:**    SVEA finds that heavy data augmentation introduces additional high variance to agent training, which can lead to instability or even divergence. SVEA suggests that using the Q-values of non-augmented images as the target of estimated Q-values for augmented images (Eq 4), thus stabilizing the variance of the value estimation.

$$\left\|Q_\theta\left(aug(x_t),\mathbf{a}_t\right) - q_t^{\text{tgt}}\right\|_2^2 \quad (4)$$

**SRM:**    SRM proposes a novel data augmentation method that operates in the frequency domain. It helps diversify data and alleviate distribution shift issues under various visual scenarios. During the training stage, SRM randomly discards parts of the frequency information from observations, forcing the policy to select suitable actions based on the remaining information. The augmentation method is shown in Eq 5, where $\mathcal{F}$ is the fast Fourier transform, $\mathbf{M}$ is a binary masking matrix, and $\mathbf{Z}$ is a random noise image.

$$\hat{\mathcal{F}}\left(o_i\right) = \mathbf{M} \cdot \mathcal{F}\left(o_i\right) + \left(\mathbf{1} - \mathbf{M}\right) \cdot \mathcal{F}(\mathbf{Z}) \quad (5)$$

**SGQN:**    This algorithm introduces the saliency map for the use of augmenting images. Saliency maps, a tool used in computer vision, offers an interpretability analysis of encoders. SGQN retains only agent's focusing areas and removes the visual background by the generated saliency map.

This approach utilizes the augmentation objectives in SVEA [23] to further improve the model's generalization performance. The auxiliary objective is shown in Eq 6, where $M_\rho((o, a), a)$ is the binary masking matrix introduced from the saliency map.

$$L_C(\theta) = \|Q_\theta(o, a) - Q_\theta(o \odot M_\rho(o, a), a)\|^2 \tag{6}$$

## B    Implementation Details

**Indoor navigation:**    Habitat serves as the simulator and extends a variety of indoor navigation tasks. We select *ImageNav* as the test env, whose goal is defined by the image of target location in the chosen map. Due to the complexity in the default training and validation episode settings, which demands extensive training periods to achieve a satisfactory standard, we simplify the setup to 500 initial positions and 1 target position. Meanwhile, we utilize the 3D scenes from the Gibson dataset as our map for all experiments.

**Autonomous driving:**    We choose the stable version of CARLA 0.9.10 for simulation. The reward function is adopted from Zhang et al. [62]. We also implement the wrapping methods from Huang et al. [24] for novel CARLA environments. Moreover, to enhance exploration and ensure stable training, we standardized the *std_schedule* across all algorithms. Each difficulty level contains two weathers, *Easy level:* soft_high_light, soft_noisy_low_light; *Medium level:* HardRainSunset, SoftRainSunset; *Hard level:* hard_low_light, hard_noisy_low_light. The aggregated return is calculated by averaging over the weather at the same level. Further details can be accessed in the documentation provided within our GitHub repository.

**Dexterous manipulation:**    In RL-ViGen, we select three single-view Adroit tasks. Given that tasks in Adroit typically necessitate demonstrations for successful completion, we employ VRL3, the state-of-the-art baseline for these tasks. Since the update process of VRL3 is based on DrQ-v2, it allows a seamless transfer of our algorithms to VRL3's codebase. There are three stages for VRL3 training: stage1 responses to gain a basic perception ability via pretraining on ImageNet; stage2 utilizes offline RL training with expert demonstrations; stage3 executes online training. It is noteworthy to mention that the experiment of VRL3 demonstrates that in single view tasks, only applying stage3 is sufficient to accomplish Adroit tasks with high sample efficiency. Therefore, to compare each algorithm more effectively, we exclude the use of stage1 and stage2. The aggregated success rate is calculated by averaging over all three tasks.

**Table-top manipulation:**    SECANT [12] previously employed Robosuite for generalization testing. Building on its codebase, we adopt one of the latest versions - Robosuite 1.4.0 and mujoco 2.3.0 as well as simplified the installation process. Meanwhile, we also introduce a range of new classes of visual generalization. For each difficulty level, we deploy a variety of scenarios, and each trained agent is evaluated within each environment 10 times (in a total of 100 evaluations). The aggregated return is calculated by averaging over all three tasks.

**Locomotion:**    In addition to the locomotion tasks from DM-Control (1.0.8 version), we also incorporate models from Mujocoreie [7], and carefully designed corresponding rewards, enabling them to accomplish *walk* or *stand* tasks. Furthermore, building on DMC-GB, we have added additional generalization categories for further enriching RL-ViGen. The aggregated return is calculated by averaging over two tasks.

Our experiments are all conducted with TeslaA40 or TeslaA100 GPU and AMD EPYC 7542 32-Core Processor CPU. More details can be founded in `https://github.com/gemcollector/RL-ViGen`.

## C    Hyper-parameters

We use the same hyper-parameters as the original papers and perform a small-scale grid search to achieve better performance of certain algorithms. The common hyper-parameters are listed in Table 2.

The individual hyper-parameters are listed in the following Tables. The additional hyper-parameters introduced by SGQN are listed as well.

Table 2: Common hyper-parameters in RL-ViGen.

| Hyper-parameters | Value |
|---|---|
| Input size | $84 \times 84$ |
| Discount factor $\gamma$ | 0.99 |
| Replay Buffer size | int(1e7) |
| Feature dim | DrQ(v2), CURL: 50, otherwise: 256 |
| Action repeat | Robosuite: 1, otherwise: 2 |
| N-step return | DrQ: 1, otherwise: 3 |
| Optimizer | Adam |
| Hidden dim | 1024 |
| Frame stack | 3 |

Table 3: CARLA hyper-parameters in RL-ViGen.

| Hyper-parameters | Value |
|---|---|
| Training Frames | int(1e6) |
| Learning Rate | PIE-G: 5e-5, DrQ: 5e-4, otherwise: 1e-4 |
| N-step return | 1 |
| SGQN quantile | 0.9 |
| SGQN critic weight | 0.5 |
| SGQN aux lr | 8e-5 |

# D   Visualization of each difficulty level

To gain a better understanding of our setting and RL-ViGen, we visualize the images under various generalization settings and difficulty levels as mentioned in the experiment section.

## D.1   Visual Appearances and Lighting Changes

### D.1.1   Robosuite

In the context of Robosuite, each difficulty level comprises 10 distinct scenes. We perform 10 trials for each of these scenes (100 trials in total).

The *Easy* level includes changes of the background appearance, while the *Hard* level contains additional complexities of moving light and alterations to the robotic arm's color. The *Extreme* level further employs a dynamic video background to evaluate the trained agents' generalization abilities. The visualized figures are shown in Figure 10.

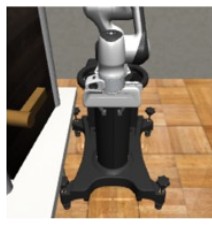
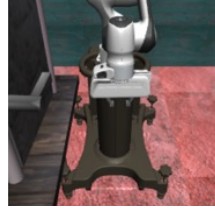
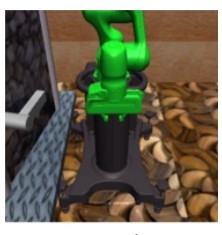
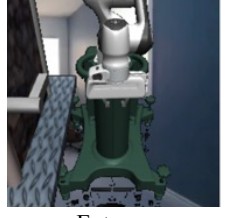

Training          Easy          Hard          Extreme

Figure 10: **The visualization of various difficulty levels of Robosuite.** This figure shows examples from the *Door* task. As the difficulty level increases, more types of distracting factors are introduced.

### D.1.2   CARLA

Apart from the default weather settings in CARLA, we implement a series of challenging new scenarios. In our CARLA setup, each level of difficulty is characterized by two specific weather conditions. The *Easy* level includes *soft_noisy_low_light* and *soft_high_light*, while the *Medium*

Table 4: Habitat hyper-parameters in RL-ViGen.

| Hyper-parameters | Value |
|---|---|
| Training Frames | int(7e5) |
| Learning Rate | 1e-4 |
| N-step return | 1 |
| SGQN quantile | 0.93 |
| SGQN critic weight | 0.9 |
| SGQN aux lr | 8e-5 |

Table 5: Adroit hyper-parameters in RL-ViGen.

| Hyper-parameters | Task | Value |
|---|---|---|
| Training Frames | Hammer | int(1e6) |
| | Door | int(1e6) |
| | Pen | int(2e6) |
| Learning Rate | Hammer | 1e-4 |
| | Door | 1e-4 |
| | Pen | 1e-4 |
| SGQN quantile | Hammer | 0.9 |
| | Door | 0.9 |
| | Pen | 0.9 |
| SGQN critic weight | Hammer | 0.9 |
| | Door | 0.5 |
| | Pen | 0.9 |
| SGQN aux lr | Hammer | 8e-5 |
| | Door | 8e-5 |
| | Pen | 8e-5 |

level is defined by the *HardRainSunset* and *SoftRainSunset* conditions. The *Hard* level contains *hard_low_light* and *hard_noisy_low_light*. As the disparity between the novel scenarios and the training images increases, the difficulty level of generalization grows. RL-ViGen also encompasses challenging conditions such as rainy, overcast, and slippery road surfaces. The visualized figures are shown in Figure 11.

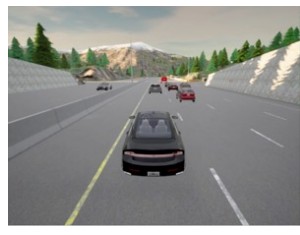
Easy
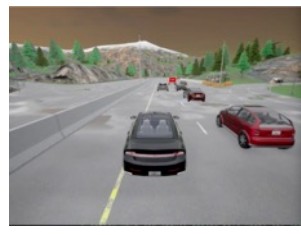
Medium
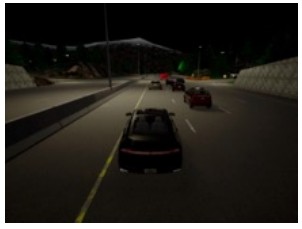
Hard

Figure 11: **The visualization of various difficult level of CARLA.** The higher the disparity from the training observations, the more challenging the new scenario.

### D.1.3 Habitat

For Habitat, the Gestaltor 3D model editor is applied to modify the appearance of the scene's 3D models. A total of 10 distinct scenarios are created. The visualized figures are shown in Figure 12.

Table 6: Robosuite hyper-parameters in RL-ViGen.

| Hyper-parameters | Task | Value |
|---|---|---|
| Training Frames | Door | int(6e5) |
| | Lift | int(8e5) |
| | TwoArmPegInhole | int(8e5) |
| Learning Rate | Door | 1e-4 |
| | Lift | DrQ(v2), CURL: 1e-4, otherwise: 8e-5 |
| | TwoArmPegInhole | SGQN: 1e-4, otherise: 8e-5 |
| Level | Door | Easy |
| | Lift | Medium |
| | TwoArmPegInhole | Medium |
| SGQN quantile | Door | 0.9 |
| | Lift | 0.9 |
| | TwoArmPegInhole | 0.87 |
| SGQN critic weight | Door | 0.7 |
| | Lift | 0.7 |
| | TwoArmPegInhole | 0.7 |
| SGQN aux lr | Door | 8e-5 |
| | Lift | 8e-5 |
| | TwoArmPegInhole | 8e-5 |

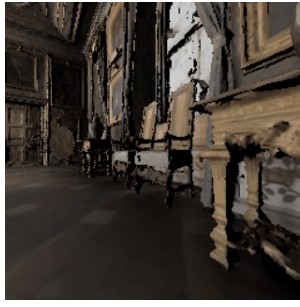 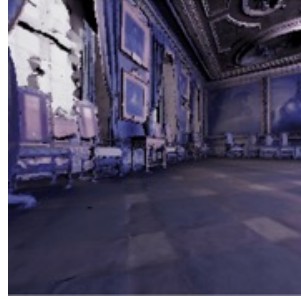 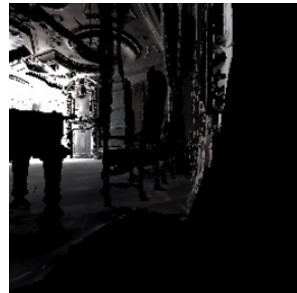

Figure 12: **The visualization of Habitat.** We create 10 distinct scenarios for the generalization of visual appearances.

### D.1.4 Locomotion

For DM-Control, we further augment numerous new types of generalizations on the basis of DMC-GB. For the unitree series tasks, *Easy* and *Hard* denote two levels of difficulty regarding light color, light position, changes of light's movement and objects' color. The visualized figures are shown in Figure 13.

### D.1.5 Adroit

In the Adroit environment, we provide four generalization scenarios. The *Color* setting changes the background, object color, and table texture, while the *Video* setting utilizes a dynamic background and introduces moving light. As illustrated in Figure 14, each scenario is configured with two levels of difficulty.

### D.2 Camera Views

For camera view generalization, we implement alternations to the camera view through the modification of camera's orientation, position, and FOV. The visualized figure is shown in Figure 15.

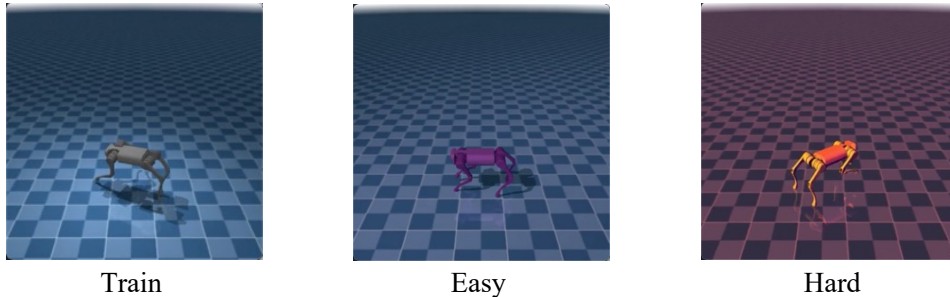

| Train | Easy | Hard |

Figure 13: **The visualization of various difficulty level of DM-Control.** The figure above show examples from unitree tasks. Factors such as light color, light position, movement of light, and object color are varied.

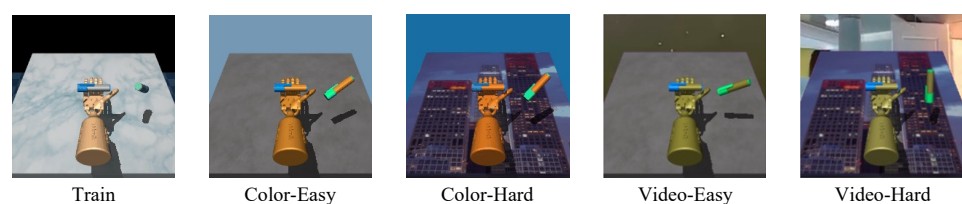

| Train | Color-Easy | Color-Hard | Video-Easy | Video-Hard |

Figure 14: **The visualization of various difficulty levels of Adroit.** This figure demonstrates examples from the *Pen* task. We show four generalization scenarios provided in RL-ViGen.

## D.3  Scene Structures

As shown in Figure 16, we established a variety of road scenarios in CARLA, including roundabouts, narrow paths, tunnels, etc., which can be also utilized in conjunction with other adjustable parameters. As the experiment illustrated in Figure 7, we employ the same weather conditions as those during training.

## D.4  Cross Embodiments

In terms of cross-embodiment generalization, we modify the type of the robotic arm in Robosuite. In addtion, by leveraging the OSC_POSE control method, the input actions are interpreted as delta values from the current state, thus facilitating to maintain the action space dimensions and corresponding meanings.

## E  Visualization of each difficulty level

To gain a better understanding of our setting and RL-ViGen, we visualize the images under various generalization settings and difficulty levels as mentioned in the experiment section.

At first, we will structure the general modifications made to each generalization type. A more detailed configuration of each environment will be specified subsequently.

- Visual appearances: The generalization type of visual appearances are divided into two variations: static and dynamic changes.
  - Color: Drawing on DMC-GB [22], we categorize color variations into two levels (*easy* and *hard*) of difficulty, each containing 100 color combinations. In both levels, alterations are made to three environmental attributes: *body*, *grid*, and *skybox*. In the *easy* level, the contrast between the color combinations and the original training scene's color combinations is 0.07, while in the *hard* level, the contrast increases to 0.14.
  - Video: We divide video variations into two levels of difficulty by replacing the environment's *skybox* with videos. The *easy* level contains 10 videos, while the *hard* level

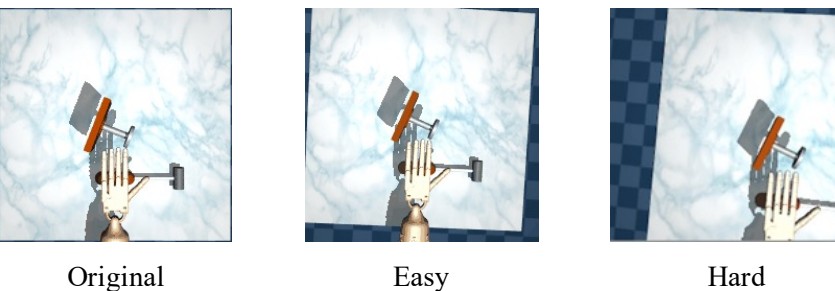

| Original | Easy | Hard |

Figure 15: **The visualization of camera views of Adroit.** The larger the deviation angle of the camera, the higher the difficulty of generalization.

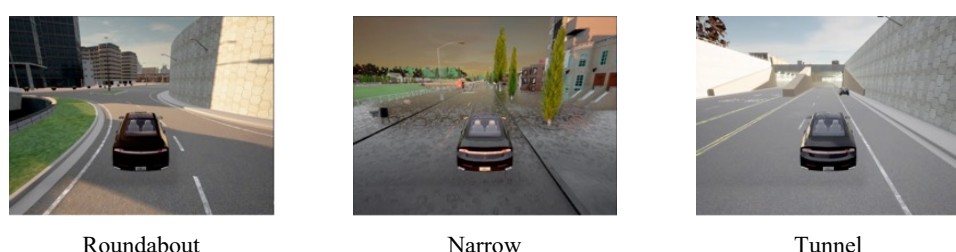

| Roundabout | Narrow | Tunnel |

Figure 16: **The visualization of scene structures of CARLA.** We selected certain locations within different maps to serve as scenarios for scene structure generalization.

consists of 100 videos with increased contrast. Additionally, the hard level also remove the reference plane of the ground.

- Lighting conditions: The generalization type of lighting conditions are divided into two variations: static and dynamic changes. Each variations are categorized into two difficulty levels.
  - Static: We adjust the static lighting conditions by altering the position, intensity, and color.
    * Position: *Easy*: the coordinates $x,y$ will be determined by adding values randomly sampled from the interval [-0.5, 0.5] via uniform sampling to the original light source coordinates, and the $z$ coordinate will be selected from the range [-0.2, 0.2]; *Hard*: the coordinates $x,y$ will be determined by adding values randomly sampled from the interval [-1, 1] via uniform sampling to the original light source coordinates, and the $z$ coordinate will be selected from the range [-0.5, 0.5].
    * Intensity: *Easy:* the value of intensity will be uniformly sampled from the interval [0.7, 1.4]; *Easy:* the value of intensity will be uniformly sampled from the interval [1.4, 2.5].
    * Color: *Easy:* the RGB values will be determined by adding values randomly sampled from the interval [-0.1, 0.1] via uniform sampling to each channel; *Hard:* the RGB values will be determined by adding values randomly sampled from the interval [-0.2, 0.2] via uniform sampling to each channel. It should be noted that the RGB values are normalized to [0, 1].
  - Dynamic: The alteration of y coordinate is defined by adding a value within the interval [-1.2, 1.2] to the original coordinate value, with an incremental change of 0.04 at each step.

- Camera views: The generalization type of camera views are divided into two variations: static and dynamic changes. Each variations are categorized into two difficulty levels.
  - Position: *Easy*: the coordinates $x,y,z$ will be determined by adding values randomly sampled from the interval [-0.03, 0.03] via uniform sampling to the original camera positions; *Hard*: the coordinates $x,y,z$ will be determined by adding values randomly

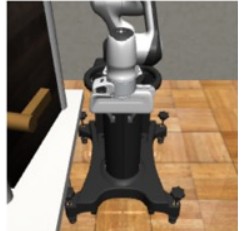 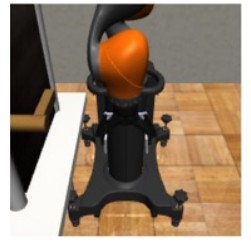 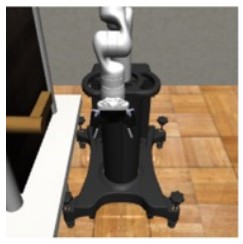

| Panda | IIWA | Kinova3 |

Figure 17: **The visualization of cross embodiment of Robosuite.** This figure shows examples from the *Door* task. Here, we demonstrate our modification of the style of the robotic arm for cross-embodiment generalization.

        sampled from the interval [-0.07, 0.07] via uniform sampling to the original camera positions.

– Orientation: *Easy*: each component of quaternion will be determined by adding values randomly sampled from the interval [-0.03, 0.03] via uniform sampling to the original camera quaternion; *Hard*: each component of quaternion will be determined by adding values randomly sampled from the interval [-0.08, 0.08] via uniform sampling to the original camera quaternion.

– FOV: *Easy*: the coordinates $x,y,z$ will be determined by adding values randomly sampled from the interval [-5, 5] via uniform sampling to the original FOV; *Hard*: the coordinates $x,y,z$ will be determined by adding values randomly sampled from the interval [-10, 10] via uniform sampling to the original FOV.

Due to the challenges in defining the difficulties associated with the generalization types of the scene structures and cross embodiments, and the complexity in numerically describing the differences between scenes, we don't provide additional quantification to these two types.

## F    Additional Results

### F.1    Generalization Evaluation

#### F.1.1    Locomotion

Built upon DM-Control, which has included numerous locomotion tasks, we extend this benchmark by integrating real-world robot models from Mujocoreie [7] with corresponding tasks. Moreover, RL-ViGen also augments DMC-GB with more tasks and generalization types. Here we evaluate the performance of each algorithm on the Unitree series tasks. Figure 18 demonstrates that all generalization algorithms exhibit comparable performance. More specifically, SVEA outperforms other techniques in the *Easy* setting, where the other generalization techniques do not show any advantages. In the *Hard* setting, where the agent's color closely resembles that of the surrounding environment, SGQN may not effectively capture the agent's outline, leading to a performance decline.

#### F.1.2    Table-top Manipulation

In Robosuite, three tasks, including single-arm and dual-arm settings, are selected in RL-ViGen: *Door*, *Lift*, and *TwoArmPegInhole*. Additionally, we create multiple difficulty levels, incorporating various visual scenarios, and dynamic backgrounds. In the *Easy* and *Medium* test environments, where considerable variations in visual colors and lighting changes are introduced, the results in Figure 19 show that PIE-G demonstrates slightly better performance than that of SGQN and SRM in *Easy* and *Medium* settings. However, when faced with the *Hard* setting that integrates dynamic video backgrounds, SRM, which mainly resorts to static frequency-based augmentation, is unable to adapt effectively to such scenarios for completing the manipulation tasks. Figure 19 further indicates that the remaining algorithms struggle to demonstrate generalization abilities in this environment.

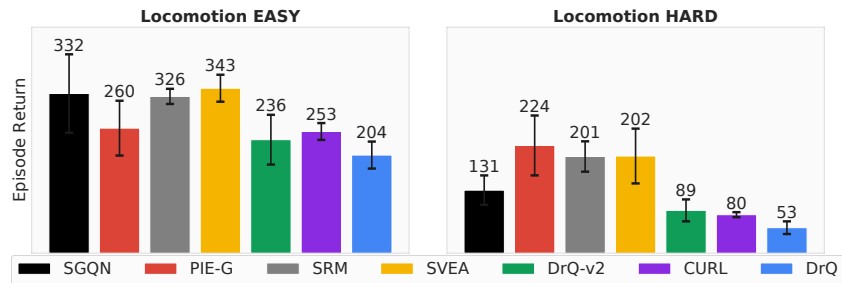

Figure 18: **Generalization score of Locomotion.** The generalization algorithms show comparable performance at two difficulty levels.

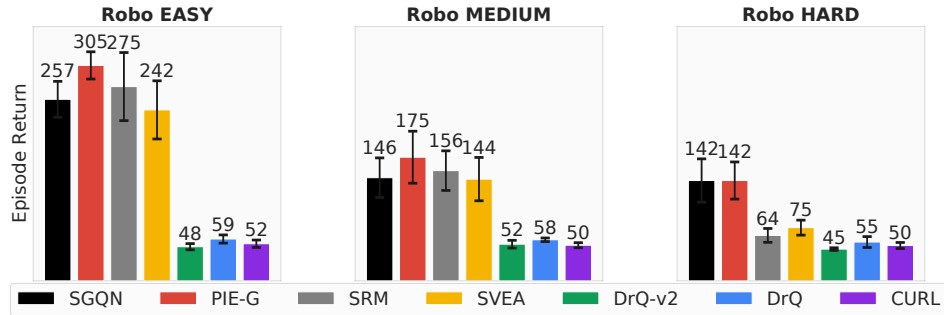

Figure 19: **The aggregated generalization score of table-top manipulation.** We present the aggregated return of three tasks for each method. PIE-G shows better generalization performance of table-top manipulation tasks when facing unseen visual scenarios.

## F.2 Wall Time

So far, our main focus has been the comparison of generalization performance of each method across various tasks. In this section, we turn our attention to the comparison of each algorithm's wall-clock training time. We choose *Walker walk* task from DMControl for evaluation. This task requires a large batch size for training, thus is suitable for better demonstrating the wall-time efficiency of each approach. Frames-per-second (FPS) is selected to be the evaluation metric. Figure 20 illustrates that DrQ-v2 owns the least computational cost. Conversely, for the algorithms that utilize additional data for augmentation purposes, they tend to exhibit lower frames-per-second (FPS) rates. SGQN builds the saliency maps during every training step, which takes extra costs. Meanwhile, PIE-G utilizes the ImageNet pre-trained ResNet model to convert high-dimensional images into representations, thus adding more burden on the model's inference compared to other algorithms.

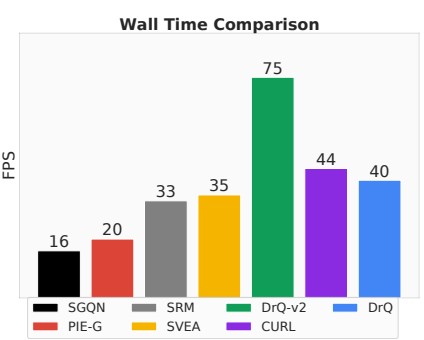

Figure 20: **Wall Time Comparison.** DrQ-v2 enjoys the lowest computational cost.

## F.3 The re-implementation of CURL

CURL [33], which adopts contrastive loss as an auxiliary objective, is frequently mentioned in numerous works [57, 32, 35], yet the effectiveness of contrastive loss appears to be less pronounced [35, 36]. Distinct from prior studies, we do not utilize a target encoder and remove the update of momentum parameters related to the encoder. As shown in Figure 21, comparing to the state-of-the-art approach DrQ-v2 and the results reported in previous work [57], the use of a single shared encoder for achieving representations seems to yield more favorable results when leveraging contrastive loss.

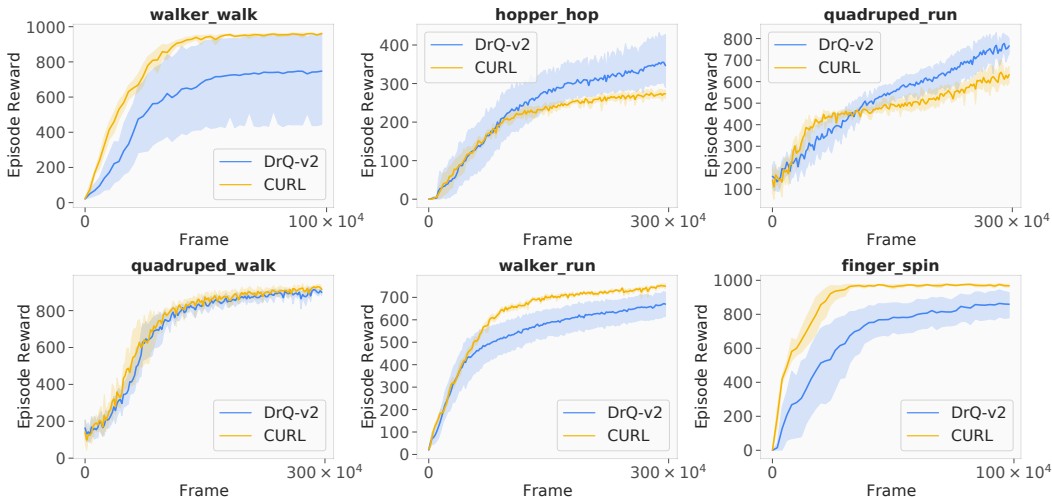

Figure 21: **The sample efficiency comparison between CURL and DrQ-v2.** Our re-implementation of CURL can achieve comparable sample efficiency with DrQ-v2.

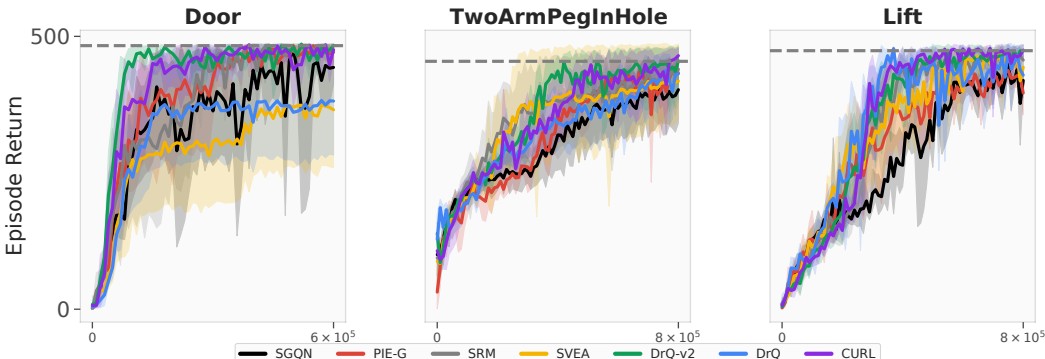

Figure 22: **Sample efficiency of Robosuite.** The episode return of each algorithm. We normalize the training steps into (0, 1). DrQ-v2 and CURL show better sample efficiency.

## F.4  Sample efficiency

In this section, we compare the sample efficiency of various visual RL algorithms. As one of the state-of-the-art visual RL algorithms, DrQ-v2 serves as a baseline for evaluating the training performance of various algorithms across different tasks. In each figure, the convergence performance of DrQ-v2 is marked with a gray dashed line. As shown in Figure 22 and Figure 23, DrQ-v2 and CURL obtain advantageous sample efficiency in locomotion and table-top manipulation tasks. A shared attribute between these two types of tasks is that the agent is positioned at the center of observation. Hence, the additional noise introduced by data augmentation tends to exacerbate training instability.

Regarding Habitat and CARLA, as shown in Figure 24, the difference of sample efficiency across diverse algorithms is minimal. This may be attributed to the fact that both two environments employ first-person view rendered images, which makes them more robust to the extra noise. Besides, it should be noted that in CARLA, agents are required to execute fast action changes on the roads to avert collisions with surrounding vehicles. Therefore, Figure 24b demonstrates that DrQ is prone to entropy collapse, while SGQN struggles to extract salient information with many distracted factors.

In terms of Adroit, as mentioned in Section 4.1.3, the safe Q mechanism is able to endow the trained agent with robustness against noise. The sample efficiency of each algorithm is shown in Figure 5.

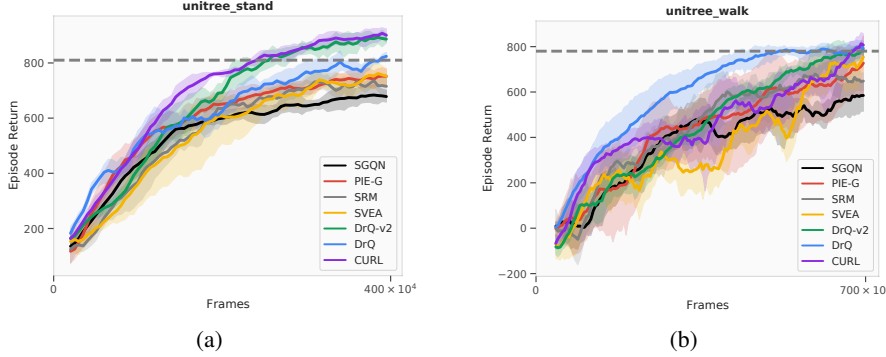

(a)                                                      (b)

Figure 23: **Sample efficiency of Unitree tasks.** The episode return of each method. The agent, positioned at the center of observation in these tasks, is subjected to additional noise due to data augmentation.

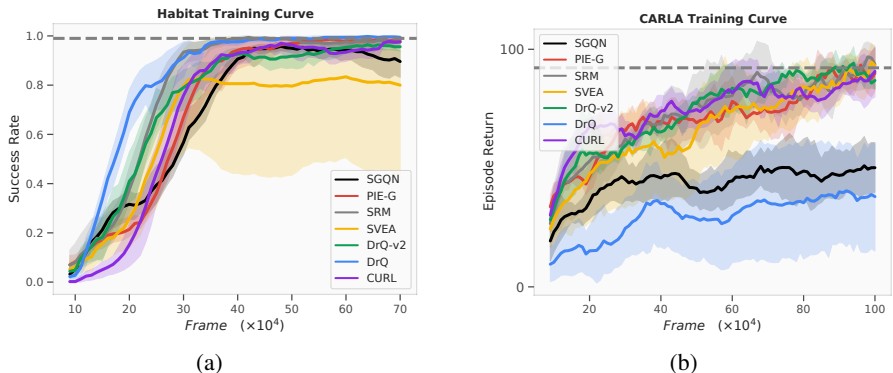

(a)                                                      (b)

Figure 24: **Sample efficiency of Habitat and CARLA.** We show the success rate of Habitat and the episode return of CARLA accordingly. The first-person view observations are more robust to the augmentation of adding additional noise.

## F.5 Training on the full-distribution scenarios

To investigate the performance of agents under test and full-distribution scenarios, we select three locomotion tasks: *walker walk*, *finger spin*, and *walker stand*. In terms of algorithmic choices, we employ three algorithms, DrQ-v2, SRM, and SVEA, to validate the training and generalization efficacy.

We have devised three distinct visual scenarios, denoted as Scenario 1, Scenario 2, and Scenario 3 for training agents and evaluating their generalization performance. Scenario 1, which serves as the training environment within the Section 4, is constructed in a static and uncluttered visual setting. Scenario 2, employed as the testing scenario for visual appearances in our work, introduces dynamic complexity by integrating video backgrounds. Scenario 3, characterized as a full distribution scenario, further amplifies this complexity by incorporating additional visual generalization types, such as changes in camera view and lighting conditions. The visualized figure are shown in Figure 25.

First, We explore the training performance of different algorithms across various visual scenarios. As shown in Figure 27, it demonstrates that as the distribution expands and the incorporated variations increase, a noticeable decline is observed in both the sample efficiency and asymptotic performance across all algorithms.

Subsequently, we investigate the generalization performance of each trained agent across three visual environments. During the generalization testing, we conduct evaluations in a zero-shot manner. The complete generalization performance are shown in Figure 28. With the increasing complexity of the testing scenarios, the generalization scores tend to decline. Moreover, the generalization scores are directly correlated with the training performance; despite the fact that Scenario 3 incorporates the

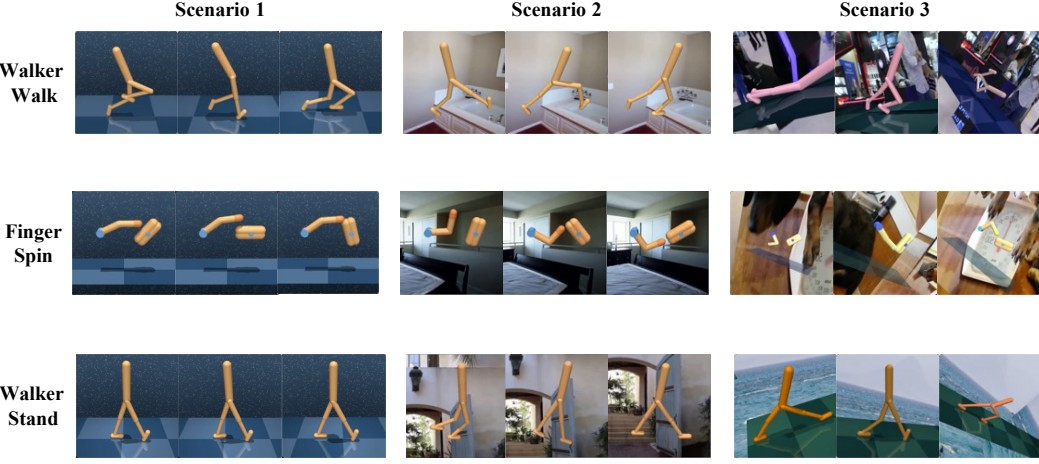

Figure 25: **Visualization of three trained tasks in different visual scenarios.** The scenario 1 is constructed in a static and clean visual scenario. Conversely, the scenario 2 introduces dynamic variations by employing a video background that alters with each episode, encompassing a total of 110 distinct videos. The scenario 3 further extends this complexity by incorporating additional visual generalization types, such as changes in camera view and lighting conditions.

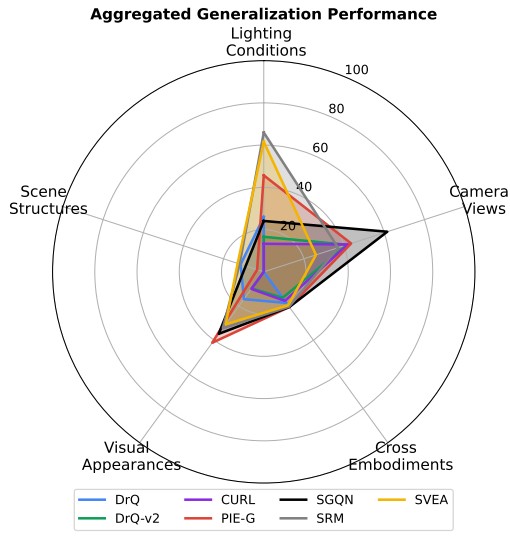

Figure 26: **Aggregated score of each algorithm.** We show the aggregated combine metric for each agent. There is no single algorithm can lead all generalization types.

most extensive array of generalization types, the inferior training performance of the agent leads to suboptimal generalization performance across various scenarios.

## F.6 The aggregated score of each algorithm

Additionally, we present an aggregated metric for each algorithm across different types of generalization. We normalize the score of each environment into 0-100, and then take the average score across all environments for each generalization type. As shown in Figure 26, no single visual RL algorithm has proven capable of adeptly handling all types of generalization, especially in the types of scene structures and cross embodiments settings. In the Discussion section, we have analyse the underlying causes of these challenges. It provides a direction for guiding future efforts towards seeking approaches that can achieve further generalization advancements in these two aspects.

## G   Additional Related Work

*Honor of Kings Arena* [52] serves as another benchmark for RL generalization. Distinct from RL-ViGen, it functions as a multi-agent platform, focusing mainly on the generalization of targets and opponents rather than visual aspects. While platforms like MineRL [18] and Malmo [26], built on Minecraft [42], are capable of handling a variety of tasks, the construction of these tasks tends to be relatively simplistic without fine-grained modeling of agents and objects. Crafter [20] and the Obstacle Tower [27], on the other hand, still utilize discrete actions, and the task types they offer are limited and lack diversity. The benchmarks such as BEHAVIOR [37] and ThreeDWorld [16] present photo-realistic environments, but their task visual scenarios are also relatively narrow and are not applicable for visual generalization evaluation.

## H   Additional Discussion

**The augmentations during training.** Both DrQ and DrQ-v2 employ data augmentation techniques like random shift or random crop. Such augmentations are referred to as weak augmentations, which only introduce minor changes to the image such as slight cropping and shifting. Numerous studies [35, 57, 32] have shown that weak augmentations are indispensable for image-based RL to achieve high sample efficiency. Absence of such weak augmentations could easily fail on most tasks [35]. However, when it comes to generalization, weak augmentations cannot help agents to obtain generalization abilities [35]. On the other hand, the augmentation method utilizing extra datasets falls into the category of strong augmentations [23, 21], which substantially distorts the image. Contrary to weak augmentations, this approach is imperative to foster superior generalization capabilities, but it will hinder agent's training performance.Therefore, most generalization algorithms utilize both types of augmentations [21, 23] to achieve generalization ability while maintaining high sample efficiency.

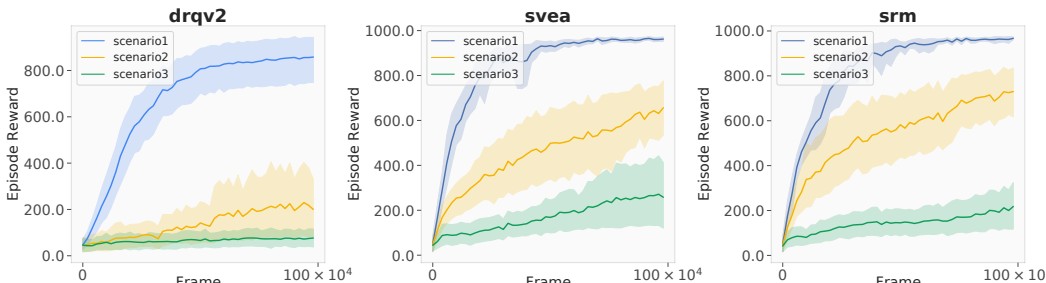

Figure 27: **Training Curves of three algorithms in different visual scenarios.** The legends are defined to represent various training curves under different scenarios. As the distribution expands and the incorporated variations increase, a noticeable decline is observed in both the sample efficiency and asymptotic performance across all algorithms.

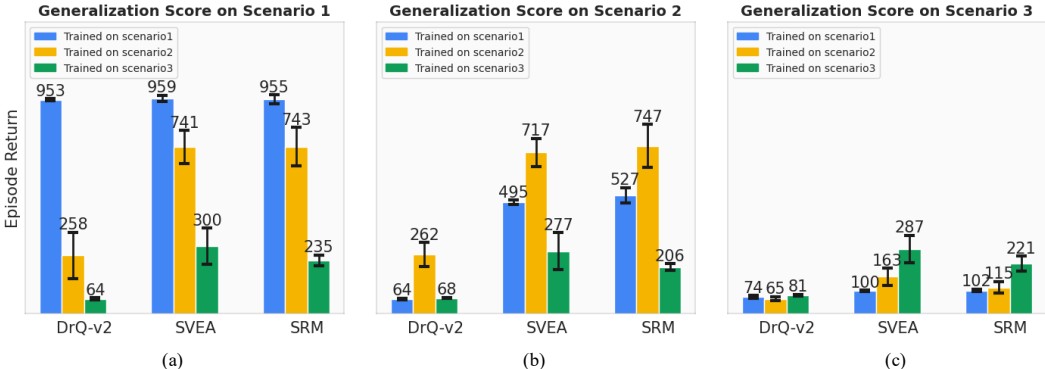

Figure 28: **Generalization performance on different visual scenarios.** In this figure, the generalization performance of each agent is evaluated across three types of visual scenarios. Specifically, the blue, yellow, and green bars represent the generalization scores evaluated in a certain scene for the agents trained under Scenario1, Scenario2 and Scenario3. Each bar represents the average performance across the three tasks.

