# OpenReview forum: "RL-ViGen: A Reinforcement Learning Benchmark for Visual Generalization"
_NeurIPS.cc/2023/Track/Datasets_and_Benchmarks — NeurIPS 2023 Datasets and Benchmarks Poster_

### Official Review · Reviewer_WGQU · 2023-07-13
**Review of RL-ViGen**

**Rating:** 7
**Confidence:** 3
**Correctness:** seems ok

**Strengths:**

- it seems that there's not a good complete benchmark for this subfield (trusting the authors didn't omit anything important in the related work), so i think they are addressing a useful problem
- implementing multiple algorithms in a unified framework is hard, and definitely can help facilitate research if the implementations are high-quality and proven to accurately reproduce the original results.
- the experimental results may be interesting to those trying to understand what the current state of the art is in this subfield
- having multiple different types of environment in one benchmark can help see the strengths and weaknesses of each algorithm

**Additional Feedback:**

none

**Clarity:**

- the abstract could be edited for improved clarity

**Documentation:**

- please make non-conda install instructions
- include videos of environments and trained agents - it's very hard to understand a RL benchmark just from images
- one of the environments needs a separate virtual environment, that seems like a pain if anyone wants to train on multiple environments simultaneously

**Ethics:**

no issues

**Limitations:**

- re-implementing RL algorithms correctly is very hard - introducing subtle mistakes or differences to the original is common and can cause differences in performance. before i feel comfortable using a new implementation of an algorithm, or trusting experimental results, i like to see proof that the re-implementation matches the original implementation. this is usually best done by running the old and new implementations on the same environments and showing they match, or you can also show that your implementation matches results from the original papers.
- each of these individual environments is not super high quality. combining a bunch of sorta crappy environments together into one benchmark feels like it just sorta masks over the issues. if you're working with one environment, you can deeply understand the limitations and work around them. but when you're using a suite like this, you kinda just treat the environments as a black box.
- in general, i'd prefer approaches that make a single high-quality environment that can encompass the various types of generalization all within the same environment. eg i am one of the authors of https://github.com/Avalon-Benchmark/avalon , which (obviously i am biased) i feel like could have been adapted easily to make an out-of-distribution-generalization benchmark (we focused on in-distribution generalization) that was more understandable and robust than the one presented here.
- installing 5 separate RL environments seems like it would be a pain from a usability perspective

**Opportunities For Improvement:**

- it wasn't immediately obvious that you were studying "out of distribution" generalization and not "in-distribution" generalization, which are very different and each are their own research field.
- it's not super clear what the training distribution is for each environment.
- sample efficiency curves need to all have an x axis. also, ideally, they'd have some horizontal lines showing the score of optimal play, or the current best published results, or something - to give context so the reader knows "are these algorithms succeeding at learning the environment or not".
- i'd like to see results of the agents trained on the full distribution, including the test scenarios. are the test scenarios even possible for these algorithms? are they struggling with out-of-distribution generalization, or with in-distribution generalization?
- i'd like to see all the individual results aggregated into a combined metric. i think a "radar chart" would be a good way to do this - group the various scenarios that each investigate a specific type of generalization (eg "lighting") together, and show an aggregate score for how each algorithm does on each type of challenge. that way future researchers can clearly and concisely show the improvements of their new methods.

**Relation To Prior Work:**

- i don't think the related work thoroughly reviewed the existing environments relevant to this research and provided a clear case for why they are insufficient. for example you say one environment has "images still being synthetic in nature", while all the environments in your work are providing synthetic images (perhaps excepting habitat for including real photos).

**Summary And Contributions:**

The authors develop a new benchmark for studying out-of-distribution generalization in visual RL environments. They collect 5 existing RL environments/simulators, and for each they add extra knobs or scenarios to create test scenarios that differ from the normal training distribution for that environment. They implemented a set of benchmarks relevant to this topic, and evaluated them. To evaluate, they trained each algorithm on each environment individually, and tested the out-of-distribution generalization using the new test scenarios for each environment.

I quite familiar with visual RL research, but not with this subfield of out-of-distribution generalization, so many of the baseline algorithms were not familiar, and I can't speak from experience to what the needs are of this subfield.

---

> ### Author Response · Authors · 2023-08-14
> **Response to Reviewer WGQU (Question 1-3)**
>
> We thank the reviewer for the time and effort you have dedicated to reviewing our work. We deeply appreciate your careful and thorough review. In the following, we seek to address each of your concerns.
>
> ---
>
> **Q1:** *" it wasn't immediately obvious that you were studying "out of distribution" generalization and not "in-distribution" generalization, which are very different and each are their own research field. The abstract could be edited for improved clarity"*
>
>
> **A:** Thank you for your inquiry. In the latest version of our paper, we will emphasize in the abstract that our benchmark is tailored for the **out-of-distribution** setting. In the field of visual generalization, the studies typically focus on visually "out of distribution" situations [1,2,3,4,5]. Moreover, it should be noted that we do not change the objectives of the task e.g., transitioning from “opening a door” to “closing a drawer”, because we do not focus on the generalization among various tasks. Our setting involves training an agent in a fixed environment and then evaluating its generalization ability in different visual scenarios. This generalization is examined in a zero-shot manner, and highlights the visual distribution shift compared to the environments in which the agents were trained. We believe your suggestion will enhance the clarity of our benchmark and make it more comprehensible to the readers.
>
> [1] Linxi Fan, Guanzhi Wang, De-An Huang, Zhiding Yu, Li Fei-Fei, Yuke Zhu, and Animashree Anandkumar. Secant: Self-expert cloning for zero-shot generalization of visual policies. In Proceedings of the 38th International Conference on Machine Learning, PMLR, 2021.
>
> [2] Nicklas Hansen, Hao Su, and Xiaolong Wang. Stabilizing deep q-learning with convnets and vision    transformers under data augmentation. Advances in Neural Information Processing Systems, 34, 2021.
>
> [3] Zhecheng Yuan et al. Don’t touch what matters: Task-aware lipschitz data augmentation for visual reinforcement learning. IJCAI 2022.
>
> [4] Liu, Siao, et al. "Improving Generalization in Visual Reinforcement Learning via Conflict-aware Gradient Agreement Augmentation." ICCV 2023.
>
> [5] Bertoin, David, et al. "Look where you look! Saliency-guided Q-networks for generalization in visual Reinforcement Learning." Advances in Neural Information Processing Systems 35 (2022): 30693-30706.
>
> ---
>
> **Q2:** *" it's not super clear what the training distribution is for each environment. Please include videos of environments and trained agents - it's very hard to understand a RL benchmark just from images"*
>
> **A:** Thanks for your suggestion. We have visualized the training scenarios used in our experiments on our website https://gemcollector.github.io/RL-ViGen/. We hope that this visualization can help you to gain a better understanding of our settings. Additionally, we would like to note that the training scenarios can be altered to match the researcher's own requirements for other settings.
>
> ---
>
> **Q3:** *" sample efficiency curves need to all have an x axis. also, ideally, they'd have some horizontal lines showing the score of optimal play, or the current best published results, or something - to give context so the reader knows "are these algorithms succeeding at learning the environment or not."*
>
> **A:**  Thank you for requesting additional details in our sample efficiency curves. We have now updated the figures to include the x-axes for all sample efficiency curves. Additionally, we use DrQ-v2 [1], which is one of the state-of-the-art visual RL algorithms, as the baseline for evaluating the training performance of various algorithms across different tasks. In each figure, the convergence performance of DrQ-v2 is marked with a gray dashed line. These figures are updated in our latest version.
>
> [1] Yarats D, Fergus R, Lazaric A, et al. Mastering Visual Continuous Control: Improved Data-Augmented Reinforcement Learning[C]//International Conference on Learning Representations. 2021.

---

> ### Author Response · Authors · 2023-08-14
> **Response to Reviewer WGQU (Question 4~5)**
>
> **Q4:** *"i'd like to see results of the agents trained on the full distribution, including the test scenarios. are the test scenarios even possible for these algorithms? are they struggling with out-of-distribution generalization, or with in-distribution generalization?."*
>
> To investigate the performance of agents under test and full-distribution scenarios mentioned in our paper, we selected three locomotion tasks: *walker walk*, *finger spin*, and *walker stand*. In terms of algorithmic choices, we employ three algorithms, DrQ-v2, SRM, and SVEA, to validate the efficacy of the experiment.
>
> We have devised three distinct visual scenarios, denoted as Scenario 1, Scenario 2, and Scenario 3 for training agents and evaluating their generalization performance. Scenario 1, which **serves as the training environment within the experimental section of our paper**, is constructed in a static and uncluttered visual setting. Scenario 2, **employed as the testing scenario for visual appearances in our work**, introduces dynamic complexity by integrating video backgrounds. Scenario 3, **characterized as a full distribution scenario**, further amplifies this complexity by incorporating additional visual generalization types, such as changes in camera view and lighting conditions. The visualized figure can be found at https://drive.google.com/file/d/14eDC1_2XaDs9GhonLSYUDUOQZLuo5eB3/view?usp=sharing .
>
> First, We explore the training performance of different algorithms across various visual scenarios. The training curve is accessible at https://drive.google.com/file/d/1cjaKr2m9sScBI5KnHVNND9lMpCYVTdXm/view?usp=drive_link . We have plotted the aggregated performance for three tasks under each visual environment. This figure demonstrates that as the distribution expands and the incorporated variations increase, a noticeable decline is observed in both the sample efficiency and asymptotic performance across all algorithms.
>
> Subsequently, we investigate the generalization performance of each trained agent across three visual environments. During the generalization testing, we conduct evaluations **in a zero-shot manner**. The complete generalization performance can be found at https://drive.google.com/file/d/1rz3hNy92n9Fe1NvhKwXNT6q9OhidFUib/view?usp=sharing. With the increasing complexity of the testing scenarios, the generalization scores tend to decline. Moreover, the generalization scores are directly correlated with the training performance; despite the fact that Scenario 3 incorporates the most extensive array of generalization types, the inferior training performance of the agent leads to suboptimal generalization performance across various scenarios.
>
>
> For RL-ViGen, the primary focus is oriented towards the challenge of training an agent within a narrow distribution environment that can exhibit the capacity to generalize to out-of-distribution unseen novel visual scenarios, rather than obtaining superior training performance in challenging visual environments. Nevertheless, as these experiments show, RL-ViGen can also be extended and applied to other domains. We hope that our benchmark can stimulate progress in other research areas.
>
> ---
>
> **Q5:** *" i'd like to see all the individual results aggregated into a combined metric. i think a "radar chart" would be a good way to do this, and show an aggregate score for how each algorithm does on each type of challenge. that way future researchers can clearly and concisely show the improvements of their new methods."*
>
> **A:**  Thanks for your suggestion. We present an aggregated metric for each algorithm across different types of generalization. The figure can be found at https://drive.google.com/file/d/1L9i40WX8P0P3XCHgsOs5eVS1N-lIjZDo/view?usp=drive_link. We normalize the score of each environment into 0-100, and then take the average score across all environments for each generalization type:
>
> $$
> agg\\_score = \\frac{1}{n}\\sum\_{i=1}\^{n}\\frac{score\_{env\_{i}}}{max\\_score\_{env\_{i}}} * 100
> $$
>
> As shown in this figure, no single current state-of-art visual RL algorithm has proven capable of adeptly handling all types of generalization. Therefore, we hope our benchmark could stimulate the advancement of agents that can truly exhibit overall visual generalization capabilities. We have added this part to the Appendix.

---

> > ### Comment · Reviewer_WGQU · 2023-08-21
> >
> > Thanks for doing this little study - these results are interesting and roughly what I would have expected (but valuable to confirm). My personal takeaway from this is - it doesn't make sense to try to get a model to generalize to a wider distribution, if learning directly on that wider distribution is already challenging for the model.
> >
> > This study seems worth including in the appendix to me, if you are confident that the results are sound.

---

> > > ### Author Response · Authors · 2023-08-27
> > > **Response to Reviewer WGQU**
> > >
> > > Thank you for your feedback. We believe your perspective will inspire the future researches in this field.  The results of full distribution training have been incorporated into the Appendix F.7. We hope that RL-ViGen can not only be beneficial for the out-of-distribution visual generalization setting but also, by offering a wide range of visual scenarios characterized by various levels, boost the training efficiency of agents when facing distracting factors in the future.

---

> ### Author Response · Authors · 2023-08-14
> **Response to Reviewer WGQU (Question 6)**
>
> **Q6:** *" re-implementing RL algorithms correctly is very hard - introducing subtle mistakes or differences to the original is common and can cause differences in performance. before i feel comfortable using a new implementation of an algorithm, or trusting experimental results, i like to see proof that the re-implementation matches the original implementation. this is usually best done by running the old and new implementations on the same environments and showing they match, or you can also show that your implementation matches results from the original papers."*
>
> **A:**  Thanks for your thoughtful suggestion! In fact, we have taken into consideration the comparison with the original code's performance and the published results during our re-implementation.
>
> In order to verify the efficacy of our re-implemented algorithm, we compare the performance of each algorithm with the results published in their corresponding original papers.
>
> **Our re-implementation has yielded competitive even better results than those achieved with the original code.** Regarding CURL [4] and DrQ [5], their original main experiments are conducted on DM-Control, and focus on sample efficiency. In contrast to prior studies, we do not utilize a target encoder and remove the update of momentum parameters related to the encoder. Consistent with the original paper, we plot sample efficiency curves of 6 tasks. The figure can be found in https://drive.google.com/file/d/1ifEBD6Xzy14pdwp-xdEj_Vpd7NC5WHYW/view?usp=sharing. As shown in this figure, compared to the state-of-the-art approach DrQ-v2 and the results reported in previous work [1] (Figure 3 and 4), our implementation yields more favorable results when leveraging contrastive loss and re-implementing DrQ in this framework. The agents trained with original code implementation fail to gain any performance in *quadruped-run*, *quadruped-walk* these 2 tasks, while our implementation can gain comparable results with the DrQ-v2.
>
>
> In terms of PIE-G [6], SGQN [7], and SVEA [8], their original experiments are conducted on DMC-GB, emphasizing generalization. As shown in this Table, our re-implementation achieves comparable performance with the results presented in the original paper on average. More results and details can be found in Appendix F.4.
> | Algorithms | published scores | our scores | $\Delta$ |
> | :-----: | :-----: | :-----: | :-----: |
> | PIE-G |  $813$   |   $818$    |  $ +5 $   |
> | SGQN |  $ 798 $    |   $ 823 $    |  $ +25 $   |
> | SVEA |  $ 842 $    |   $831 $    |  $-11$   |
>
> For the algorithms DrQ-v2 [1], VRL3 [2], and SRM [3], we employed the original code in the environments used in the corresponding original papers. This was done without any modifications to ensure reproducibility. In environments not addressed in the original studies, we conducted further parameter tuning.
>
> We added the relevant details and results to Appendix F.4.
>
> ---
>
> [1] Yarats D, Fergus R, Lazaric A, et al. Mastering Visual Continuous Control: Improved Data-Augmented Reinforcement Learning[C]//International Conference on Learning Representations. 2021. Original code repo:      https://github.com/facebookresearch/drqv2
>
> [2] Wang, Che, et al. "Vrl3: A data-driven framework for visual deep reinforcement learning." Advances in Neural Information Processing Systems 35 (2022): 32974-32988. Original code repo: https://github.com/microsoft/VRL3
>
> [3] Huang Y, Peng P, Zhao Y, et al. Spectrum Random Masking for Generalization in Image-based Reinforcement Learning[J]. Advances in Neural Information Processing Systems, 2022, 35: 20393-20406. Original code repo: https://github.com/Yara-HYR/SRM
>
> [4] Laskin M, Srinivas A, Abbeel P. Curl: Contrastive unsupervised representations for reinforcement learning[C]//International Conference on Machine Learning. PMLR, 2020: 5639-5650. Original Code repo: https://www.github.com/MishaLaskin/curl
>
> [5] Kostrikov, Ilya, Denis Yarats, and Rob Fergus. "Image augmentation is all you need: Regularizing deep reinforcement learning from pixels." arXiv preprint arXiv:2004.13649 (2020). Original code repo: https://github.com/denisyarats/drq
>
> [6] Yuan, Zhecheng, et al. "Pre-trained image encoder for generalizable visual reinforcement learning." Advances in Neural Information Processing Systems 35 (2022): 13022-13037.   Original code repo: https://github.com/gemcollector/PIE-G
>
> [7] Bertoin, David, et al. "Look where you look! Saliency-guided Q-networks for generalization in visual Reinforcement Learning." Advances in Neural Information Processing Systems 35 (2022): 30693-30706. Original code repo: https://github.com/SuReLI/SGQN
>
> [8] Nicklas Hansen, Hao Su, and Xiaolong Wang. Stabilizing deep q-learning with convnets and vision    transformers under data augmentation. Advances in Neural Information Processing Systems, 34, 2021. Original code repo: https://github.com/nicklashansen/dmcontrol-generalization-benchmark

---

> ### Author Response · Authors · 2023-08-14
> **Response to Reviewer WGQU (Question 7-8)**
>
> **Q7:** *"in general, i'd prefer approaches that make a single high-quality environment that can encompass the various types of generalization all within the same environment. eg i am one of the authors of https://github.com/Avalon-Benchmark/avalon , which (obviously i am biased) i feel like could have been adapted easily to make an out-of-distribution-generalization benchmark (we focused on in-distribution generalization) that was more understandable and robust than the one presented here."*
>
> **A:** In the real world, when facing diverse tasks, the structure of the task, the dynamics, and the forms of the controlled agents are all different. Previous algorithms have mostly been verified for their effectiveness under a single task environment and a specific type of generalization [1,2,3,4,5]. We believe that, for out-of-distribution visual generalization, this experiment setup may pose a certain risk of promoting overfitting to the benchmark, rather than discovering the algorithms that could be potentially beneficial for solving real-world problems. Therefore, we recognize the necessity to validate the effectiveness of algorithms across various environments.
>
> As demonstrated in our experiments, the algorithms exhibit distinct characteristics when faced with different environments. Furthermore, none of the algorithms that are claimed to be state-of-the-art (SOTA) are capable of mastering all the environments. This observation highlights the need for RL-ViGen to provide a comprehensive examination of algorithms in the research area of visual generalization.
>
> The difference between Avalon and RL-ViGen:
>
> - The Avalon benchmark shares a unified world dynamics and task structure, making it highly suitable as a benchmark for in-distribution generalization. In contrast, RL-ViGen focuses more on out-of-distribution scenarios, and is designed to handle multiple environments.
> - The generalization type of Avalon mainly concentrates on the task level, while RL-ViGen is concerned with the visual level. RL-ViGen investigates whether an agent can still accomplish a given task when the visual input changes, emphasizing the agent's ability to adapt within the same task under varying visual scenarios.
>
> Avalon is another valuable benchmark for RL generalization. We have now included a discussion of Avalon in the related work section of our paper (Line 310~312). Thank you once again for highlighting this valuable work.
>
>
> [1]  Nicklas Hansen and Xiaolong Wang. Generalization in reinforcement learning by soft data augmentation. In 2021 IEEE International Conference on Robotics and Automation (ICRA), pages 13611–13617. IEEE, 2021.
>
> [2] Bertoin, David, et al. "Look where you look! Saliency-guided Q-networks for generalization in visual Reinforcement Learning." Advances in Neural Information Processing Systems 35 (2022): 30693-30706.
>
> [3] Liu, Siao, et al. "Improving Generalization in Visual Reinforcement Learning via Conflict-aware Gradient Agreement Augmentation." ICCV 2023.
>
> [4] Wu K, Wu M, Chen Z, et al. Generalizing reinforcement learning through fusing self-supervised learning into intrinsic motivation[C]//Proceedings of the AAAI Conference on Artificial Intelligence. 2022, 36(8): 8683-8690.
>
> [5] Wang, Kaixin, et al. "Improving generalization in reinforcement learning with mixture regularization." Advances in Neural Information Processing Systems 33 (2020): 7968-7978.
>
> ---
>
> **Q8:** *"each of these individual environments is not super high quality. combining a bunch of sorta crappy environments together into one benchmark feels like it just sorta masks over the issues. If you're working with one environment, you can deeply understand the limitations and work around them. but when you're using a suite like this, you kinda just treat the environments as a black box."*
>
>
> **A:** I will address your inquiry from three perspectives:
> - The utilization of multiple environments: As mentioned in the Q6, employing multiple environments is vital for mitigating overfitting and demonstrating the relative pros and cons of various algorithms.
>
> - Quality in visual generalization benchmark: In comparison to typical benchmarks addressing out-of-distribution problems (mentioned in Related Work), RL-ViGen exhibits a higher level of quality.
>
>  - Flexibility and Adjustability within Environments: Our environments are not black-box scenarios. In RL-ViGen, the degree of variation for each environment can be conveniently adjusted.  We provide the config file for each environment. Users can set different difficulty levels for each generalization type. Additionally, we provide relevant interfaces that provide users the flexibility to make more sophisticated modifications according to their specific needs.

---

> ### Author Response · Authors · 2023-08-14
> **Response to Reviewer WGQU (Question 9-11)**
>
> **Q9:** *"i don't think the related work thoroughly reviewed the existing environments relevant to this research and provided a clear case for why they are insufficient. for example you say one environment has "images still being synthetic in nature", while all the environments in your work are providing synthetic images (perhaps excepting habitat for including real photos)."*
>
> **A:** We have revised the expressions in our manuscript and have expanded our comparison with existing benchmarks. These updates and detailed comparisons can be found in Appendix H and Section 6 Related work.
>
> ---
>
> **Q10:** *"installing 5 separate RL environments seems like it would be a pain from a usability perspective."*
>
>
> **A:** Thanks for your suggestion. We have made updates to address this issue. By selecting packages that are compatible across all environments, we have been able to create a single conda environment that is suitable for all tasks.
>
> ---
>
> **Q11:** *"please make non-conda install instructions"*
>
>
> **A:**  Thanks for your recommendation. We have additionally provided an installation method using Docker in our codebase.

---

> ### Comment · Reviewer_WGQU · 2023-08-21
> **response**
>
> Thanks to the authors for the detailed response and improvements to your paper. Many of my concerns were addressed and I think the paper/benchmark is more compelling now. I have increased my rating accordingly.

---

### Official Review · Reviewer_jzXK · 2023-07-20
**Review for Visual RL benchmarks**

**Rating:** 6
**Confidence:** 4
**Correctness:** 1. In Experiment 4.2, the agent that …
**Clarity:** It's well written and easy to underst…

**Strengths:**

1. Research on reinforcement learning to process vision input, which is a high-dimensional observation, is a time-consuming work, and especially reinforcement learning that is parameter sensitive is not easy to learn successfully. Under these circumstances, I believe that experiments on various tasks and their generalization would be beneficial to the RL community.
2. As written in the above contributions, I believe that the variety of generalization types and tasks is a key strength of this study.

**Additional Feedback:**

I believe this is a study that can be widely utilized, and future work is promising, so I will raise the score if my main concerns are addressed.

**Documentation:**

Necessary documentation is well organized and available via github.

**Ethics:**

No ethical issues

**Limitations:**

1. I agree that cross embodiment is an important category to guide future work, especially in the long term, but I question whether it is in the right topic range as a visual RL benchmark for generalization.
2. Overall, the study is good for experimenting in different settings, but needs to reconsider whether the experiments were appropriate for each task.

**Opportunities For Improvement:**

1. It would be great if you could also suggest a quantifiable or structured criteria for the difficulty level of Easy, Medium, and Hard. For example, the number of colors or the degree of contrast.
2. It is necessary to share the parameters of the learning environment such as reward design and max step, etc.
3. I thought Figure 24 in appendix would be the learning curve of figure3 and 4, but why do the results look different?

**Relation To Prior Work:**

A wider variety of related work may be considered in this paper.
[1] Kim et al. Goal-aware cross-entropy for multi-target reinforcement learning. NeurIPS, 2021.
[2] Yang et al. Towards Applicable Reinforcement Learning: Improving the Generalization and Sample Efficiency with Policy Ensemble. IJCAI 22
[3] Albrecht et al. Avalon: A Benchmark for RL Generalization Using Procedurally Generated Worlds. NeurIPS, 2022.

**Summary And Contributions:**

This paper proposed a benchmark that considers generalization as one of the key issues in Visual RL, and reflects 4-5 generalization types (Visual appearance, Camera views, etc.) for 5 kinds of tasks (autonomous driving, dexterous manipulations, etc.). As an important contribution point, it is common to find papers that consider the degree of visual appearance, but it can be said that the core contribution of this paper is that it considers various generalization types such as camera view and lighting. In addition, there are few benchmarks that reflect each generalization type for various tasks such as dexterous manipulation, driving, and navigation tasks. Finally, by providing benchmarks and baselines in a unified framework, it is expected to be highly utilized in the future.

---

> ### Author Response · Authors · 2023-08-14
> **Response to Reviewer jzXK (Question 1)**
>
> We thank the reviewer for the time and effort you have dedicated to reviewing our work. We deeply appreciate your careful and thorough review. In the following, we seek to address each of your concerns.
>
> **Q1:** *"I agree that cross embodiment is an important category to guide future work, especially in the long term, but I question whether it is in the right topic range as a visual RL benchmark for generalization. Cross embodiments experiments also need to be reconsidered in terms of design or comparison model."*
>
> **A:** The cross embodiment also stands as an important generalization category for visuo-motor control.  We will address your concern from three aspects: the related work, the types of cross embodiment, and the experiment.
> For the related work, many works [1,2,3] rely on visual information to accomplish cross embodiment tasks in image-based decision-making tasks. It is noteworthy that recent works, including papers such as Zhao et al. [4], also align with our perspective, treating cross embodiment (mentioned as morphology) as a form of generalization. Therefore, cross embodiment is widely recognized as an accepted category of visual generalization.
>
> For the types of cross embodiment, it can be categorized by whether owning a shared action space. In RL-ViGen, both table-top manipulation and autonomous driving belong to the category of sharing action space. Regarding table-top manipulation, as detailed in our experiments (Section 4.4), the actions of different robot arms can be uniformly defined by a 7-dimensional vector indicating the pose of the end effector. This is the video result of executing the same action under different embodiments：https://drive.google.com/drive/folders/1_asxsHQPbwiBktLsGbeR2zUq-bEz6W6U?usp=sharing. As for autonomous driving in our benchmark, different vehicles share the same action space, consisting of two continuous controls, thrusting and steering.  In terms of locomotion and dexterous hand manipulation, due to various types of robots and robotic hands, it becomes infeasible to maintain a shared action space.
>
> For the visual environments that share the same action space, since the policy outputs in these scenarios have identical semantics, the challenge of cross-embodiment lies exactly in visual understanding. This type of cross embodiment allows for a direct evaluation across different scenarios in our visual generalization setting.
> Therefore, cross-embodiment is one of the crucial visual  generalization categories worth investigating.
>
> Furthermore, our experiments demonstrate that current visual generalization RL algorithms tend to exhibit relatively suboptimal performance with respect to this category of generalization. However, the algorithms designed for visual generalization outperform those mainly focused on sample efficiency. **This result not only highlights the benefit of visual generalization enhancements for cross embodiments, but also emphasizes that this type of generalization is distinct from visual appearances or lighting conditions** (the algorithms exhibit a more robust generalization ability in these two types). Thus, it reaffirms cross embodiments as a more challenging and important visual RL generalization problem.
>
> For the visual environments where a shared action space is absent,  the visual generalization algorithms are incapable of achieving direct zero-shot transfer for generalization evaluation. However, the inability to achieve zero-shot generalization is an issue at the action level. This does not undermine the importance of visual understanding in cross-embodiment; it remains a challenging aspect of visual RL generalization.
>
>
> What’s more, compared to the cross-embodiment environments employed in prior papers [1,3], our benchmark offers tasks and environments that are much more diverse and challenging. More sophisticated modeling of objects is implemented to close the real-world conditions in RL-ViGen.
>
>
>
> [1] Zhang, Qiang, et al. "Learning Cross-Domain Correspondence for Control with Dynamics Cycle-Consistency." International Conference on Learning Representations. 2020.
>
> [2] Xu, Mengda, et al. "XSkill: Cross Embodiment Skill Discovery." arXiv preprint arXiv:2307.09955 (2023).
>
> [3] Zakka, Kevin, et al. "Xirl: Cross-embodiment inverse reinforcement learning." Conference on Robot Learning. PMLR, 2022.
>
> [4] Tony Z Zhao, Siddharth Karamcheti, Thomas Kollar, Chelsea Finn, Percy Liang. What Makes Representation Learning from Videos Hard for Control?
>
> [5] Nakanishi J, Cory R, Mistry M, et al. Operational space control: A theoretical and empirical comparison[J]. The International Journal of Robotics Research, 2008, 27(6): 737-757.

---

> ### Author Response · Authors · 2023-08-14
> **Response to Reviewer jzXK (Question 2-4)**
>
> **Q2:** *"Overall, the study is good for experimenting in different settings, but needs to reconsider whether the experiments were appropriate for each task. In Experiment 4.2, the agent that learned only on highway was evaluated with different tunnels and weather conditions, and I was wondering why you chose baseline for that experimental design. In my opinion, meta learning should be selected as the baseline, or various conditions should be considered from the training env."*
>
>  **A:** In the study of visual generalization tasks, such an experimental design is prevalent [1,2,3,4,5]. This setting involves training an agent in a single environment and testing its generalization ability in different visual environments in a zero-shot manner. The training environment is typically constructed within a static and uncluttered visual setting, while the testing environments introduce substantial visual distribution shifts. On our website, we have included GIFs of both the examples of training and testing scenarios. We believe these GIFs will provide you with a deeper understanding of our experiment design.
>
> Our benchmark serves this specific setting, thus necessitating the construction of numerous test scenarios that are visually different from the training environment. This setting is widely adopted in this research area, often incorporating numerous visual variations between the training and testing environments to challenge the agent's visual generalization abilities. **Our goal is to endow  agents the ability to generalize to out-of-distribution visual scenarios, rather than obtaining superior training performance in challenging visual environments.**
>
> In terms of algorithms, we selected the most cutting-edge visual generalization algorithms proposed in recent years, as well as classic visual reinforcement learning approaches. According to the experimental results from RL-ViGen, we note that, as of now, there are no existing generalization algorithms that can adeptly manage all tasks and generalization types. Prior existing benchmarks are insufficient to showcase this point. Therefore, we hope that our benchmark could promote the advancement of agents that can truly exhibit overall visual generalization capabilities.
>
> In the aspect of environments, meta-learning environments focus more on task-level variations, while our benchmark is primarily tailored for visual-level changes. Nevertheless, RL-ViGen can also be extended and applied to other domains. We hope that our benchmark can stimulate progress in other research areas.
>
>
> [1] ​​Linxi Fan, Guanzhi Wang, De-An Huang, Zhiding Yu, Li Fei-Fei, Yuke Zhu, and Animashree Anandkumar. Secant: Self-expert cloning for zero-shot generalization of visual policies. In Proceedings of the 38th International Conference on Machine Learning, PMLR, 2021.
>
> [2] Nicklas Hansen, Hao Su, and Xiaolong Wang. Stabilizing deep q-learning with convnets and vision    transformers under data augmentation. Advances in Neural Information Processing Systems, 34, 2021.
>
> [3] Zhecheng Yuan et al. Don’t touch what matters: Task-aware lipschitz data augmentation for visual reinforcement learning. IJCAI 2022.
>
> [4] Liu, Siao, et al. "Improving Generalization in Visual Reinforcement Learning via Conflict-aware Gradient Agreement Augmentation." ICCV 2023.
>
> [5] Bertoin, David, et al. "Look where you look! Saliency-guided Q-networks for generalization in visual Reinforcement Learning." Advances in Neural Information Processing Systems 35 (2022): 30693-30706.
>
> ---
>
> **Q3:** *" I thought Figure 24 in appendix would be the learning curve of figure3 and 4, but why do the results look different."*
>
> **A:** As illustrated at the beginning of Section 4, our setting is that all agents are trained in the same fixed training environment and evaluated within various unseen scenarios in a zero-shot manner.  Figure 24 in the appendix shows the performance under the training scenarios, while Figures 3 and 4 show the agent’s performance under various testing scenarios. Therefore, it will cause a performance drop in general due to the visual distribution shift. The better the trained agent's performance in new scenarios, the stronger the algorithm's visual generalization ability. The training and testing scenarios are visualized on our website (videos), Figure 1, and Appendix E (images).
>
> ---
>
> **Q4:** *"It would be great if you could also suggest a quantifiable or structured criteria for the difficulty level of Easy, Medium, and Hard. For example, the number of colors or the degree of contrast."*
>
> **A:** Thanks for your valuable suggestion. We have elaborated on the structured criteria for the difficulty levels of generalization types in Appendix E (Line 697 to Line 750). This clarification should enhance the understanding of the distinctions in difficulty level within our benchmark. We greatly appreciate your insightful recommendation!

---

> > ### Comment · Reviewer_jzXK · 2023-08-23
> > **response**
> >
> > I think the author's work in designing and experimenting with various tasks while proposing the novel benchmark is very valuable for research. I will raise the score from 5 to 6, but I would like to see the authors supplement their experiments with a proper baselines for each of the tasks they propose.

---

> > > ### Author Response · Authors · 2023-08-27
> > > **Response to Reviewer jzXK**
> > >
> > > Thanks for your feedback. In varied settings, the appropriate baselines differ correspondingly. We have set proper baselines across three aspects: training, visual generalization settings, and other settings for each generalization type.
> > >
> > >
> > > Regarding training, a proper baseline can achieve high sample efficiency and asymptotic training performance. We use DrQ-v2 [1], which is one of the state-of-the-art visual RL algorithms, as the baseline for evaluating the training performance of various algorithms across different tasks. In each figure, the convergence performance of DrQ-v2 is marked with a gray dashed line. These figures are updated in our latest version.
> > >
> > >
> > > RL-ViGen serves the visual generalization setting. Hence, for the visual generalization setting, the proper baselines can gain strong visual generalization abilities. The algorithms we selected in RL-ViGen are the most cutting-edge visual generalization algorithms proposed in recent years. These algorithms all are considered as appropriate baselines for various tasks within this setting.
> > >
> > >
> > > In terms of other settings, we conduct experiments on two generalization types where the generalization algorithms demonstrate relatively suboptimal performance.  MoVie [2] operates within the domain adaptation setting, specifically addressing variations in camera view. It utilizes the dynamics model as the supervision to adapt encoder to different views and incorporate spatial transformer networks (STN) in the encoder for better adaptation under the view change. We select three locomotion tasks: *cup catch*, *finger spin*, and *reacher easy* for evaluating agents’ camera-view generalization abilities.
> > >
> > > It should be noted that MoVie updates model parameters when facing novel visual scenarios, while the generalization baselines are merely evaluated in a zero-shot manner.  As shown in the following table, MoVie  can adapt more effectively in the view-changing environment.
> > >
> > > | Camera Views | MoVie | PIE-G | SGQN | SVEA |
> > > | :-----: | :-----: | :-----: | :-----: | :-----: |
> > > | cup catch |  $962 \pm 3$    |  $834 \pm 23$    |  $776 \pm 124$   | $744 \pm 71$   |
> > > | finger spin |  $892 \pm 2$   |   $680 \pm 37$    | $383 \pm 7$    | $312 \pm 118 $   |
> > > | reacher easy|  $985 \pm 3 $   |   $583 \pm 259 $  |  $969 \pm 26 $   | $ 913 \pm 7 $   |
> > >
> > > Due to the absence of algorithms specifically tailored for the cross embodiment generalization type within visual RL, we integrated PIE-G [3] with SRM [4] to enhance the agent's generalization capabilities. We employed the encoder provided by PIE-G to obtain a robust visual representation and then augmented the data with SRM's frequency-based augmentation approach to increase diversity. Table-top manipulation is employed as the evaluation platform. The aggregated generalization results can be found at https://drive.google.com/file/d/1LxONBCVisZPPbDKe9eWlgxWv-6HeWRee/view?usp=sharing . As shown in the figure, PIE-G augmented with SRM exhibits slightly better performance in the cross embodiment scenario.
> > >
> > >
> > > Furthermore, we should emphasize that RL-ViGen mainly focuses on evaluating the visual generalization abilities of trained agents. Our benchmark has incorporated  the recent state-of-the-art algorithms.  We hope that our benchmark could promote the development of superior algorithms. In addition, we will remain engaged in this domain, continually integrating relevant algorithms into our benchmark.
> > >
> > >
> > >
> > > [1] Yarats D, Fergus R, Lazaric A, et al. Mastering Visual Continuous Control: Improved Data-Augmented Reinforcement Learning[C]//International Conference on Learning Representations. 2021.
> > >
> > > [2] Yang, Sizhe, Yanjie Ze, and Huazhe Xu. "MoVie: Visual Model-Based Policy Adaptation for View Generalization." arXiv preprint arXiv:2307.00972 (2023).
> > >
> > > [3] Yuan, Zhecheng, et al. "Pre-trained image encoder for generalizable visual reinforcement learning." Advances in Neural Information Processing Systems 35 (2022): 13022-13037.
> > >
> > > [4] Huang Y, Peng P, Zhao Y, et al. Spectrum Random Masking for Generalization in Image-based Reinforcement Learning[J]. Advances in Neural Information Processing Systems, 2022, 35: 20393-20406.

---

> ### Author Response · Authors · 2023-08-14
> **Response to Reviewer jzXK (Question 5-6)**
>
> **Q5:** *" It is necessary to share the parameters of the learning environment such as reward design and max step, etc."*
>
> **A:** This part was in good format in our codebase. In response to the suggestion, we have incorporated a new section, Appendix D (Line 645 ~ Line 693), in the appendix to introduce information of environment parameters, configurations, and reward designs.
>
>
> ---
>
> **Q6:** *" A wider variety of related work may be considered in this paper."*
>
> **A:** Thank you for bringing attention to these works. We have acknowledged their relevance and cited them in the latest version of the manuscript.

---

### Official Review · Reviewer_Jh47 · 2023-07-20
**Review of RL-ViGen**

**Rating:** 7
**Confidence:** 4
**Correctness:** Yes.
**Clarity:** Yes.

**Strengths:**

+ A very comprehensive benchmark on visual RL generalization with a wide range of tasks that are very relevant to robotics learning and autonomous driving;
+ The visual generalization types are in good coverage;
+ The benchmarked algorithms are state-of-the-art and the latest developments in the direction;
+ All benchmarked algorithms are (re-)implemented in a unified codebase where they could make fair comparisons across algorithms;
+ They concluded that no single algorithm has demonstrated winning performance across all evaluated tasks; also each latest algorithm in the benchmarks has its own unique strengths;

**Additional Feedback:**

NA

**Documentation:**

Yes.

**Ethics:**

No.

**Opportunities For Improvement:**

Sim2Real is yet another aspect of generalization researchers concern a lot, especially who work on robotics. Considering the background of the authors, I hope that the authors could discuss this aspect.

**Relation To Prior Work:**

Yes.

**Summary And Contributions:**

The paper benchmarks diverse tasks and wide generalization types in visual reinforcement learning. They incorporated state-of-the-art generalization algorithms into a unified codebase which facilitates fair comparisons and further development of algorithms in this direction. The tasks in the benchmark include autonomous driving, dexterous manipulation, locomotion, etc, and generalization types cover visual appearances, camera views, cross embodiments, etc.

---

> ### Author Response · Authors · 2023-08-14
> **Response to Reviewer Jh47**
>
> We thank the reviewer for the time and effort you have dedicated to reviewing our work. We deeply appreciate your careful and thorough review. In the following, we seek to address each of your concerns.
>
> **Q:** *"Sim2Real is yet another aspect of generalization researchers concern a lot, especially who work on robotics. Considering the background of the authors, I hope that the authors could discuss this aspect."*
>
> **A:** Thanks for your question. Indeed, our benchmark can be used to facilitate more effective sim2real transfer for robots in the future. The tasks selected in RL-ViGen all have practical applications and incorporate photorealistic images, which serves to narrow the visual gap between simulations and reality. Moreover, our benchmark provides a unified evaluation of current state-of-the-art visual RL algorithms, with the hope of offering feasible solutions for algorithm selection under different tasks in sim2real scenarios.
>
> We also plan to construct real-world experiment platforms in the future to validate the effectiveness of RL-ViGen in a sim2real setting. Here is an example of using our benchmark to implement a sim2real experiment: https://drive.google.com/drive/folders/1aDHArBNi3VEJrn-5PZudiWuCeKBy_TG3?usp=drive_link. As demonstrated in the video, our benchmark enables successful **zero-shot** sim2real transfer for visual RL algorithms.

---

### Official Review · Reviewer_jky4 · 2023-07-20
**A benchmark on the generalizability of reinforcement learning.**

**Rating:** 8
**Confidence:** 5
**Correctness:** Good.
**Clarity:** This paper is well written.

**Strengths:**

This paper discusses the generalization of deep RL, including building a new diverse benchmark and evaluating related deep RL algorithms, which is a very important topic for the RL community. This may inspire a new series of studies on the generalizability of reinforcement learning.

**Additional Feedback:**

Check limitations.

**Documentation:**

Good

**Ethics:**

No ethical concerns.

**Limitations:**

This is a good paper. Some minor issues should be addressed.

1. In line 163, “This implies that the trained agent only produces a single distribution of action in response to diverse image inputs.” Have you verified this?

2. DrQ and DrQ-v2 use data augmentation techniques to do representation learning. These data augmentation approaches overlap to some extent with the augmented dataset approach used in this paper. Therefore an additional discussion of this point will help the reader to understand this benchmark.

3. The new benchmark as a whole falls under the category of presenting new challenges to the generalizability of reinforcement learning from the **visual side**. How does this new benchmark benefit the RL algorithms themselves rather than how to do representation learning better?

I would be happy to raise my score if these minor issues were fixed.

**Opportunities For Improvement:**

Hope to see the authors could develop new algorithms to improve generalization based on these benchmarks.

**Relation To Prior Work:**

This paper clearly discussed how this work differs from previous benchmarks.

**Summary And Contributions:**

This paper proposes a new benchmark RL-ViGen with multiple diverse, realistic tasks, and does extensive experiments to demonstrate the distinct performance of existing approaches when tackling diverse tasks. The limitations of some generalizable visual RL methods are shown in this paper.

---

> ### Author Response · Authors · 2023-08-14
> **Response to Reviewer jky4 (Question 1)**
>
> We thank the reviewer for the detailed and thorough review. We added the suggested experiments to the rebuttal revision. In the following, we seek to address each of your concerns.
>
> ---
>
> **Q1:** *"In line 163, “This implies that the trained agent only produces a single distribution of action in response to diverse image inputs.” Have you verified this?"*
>
> **A:**  Thanks for your suggestion. We plot the action distribution of DrQ agents in autonomous driving tasks. For comparison, we choose SRM, which demonstrates the best generalization ability on CARLA as a baseline. The figure can be found at https://drive.google.com/file/d/1etxOu2TKB_QYMM3Rb1toSI0JiOdIU_t8/view?usp=sharing. The corresponding videos can be found at https://drive.google.com/drive/folders/14tNA9xCsQFvd04EBR_L4hL0o6uEIJJ3R?usp=sharing.
>
> As shown in the figure, DrQ agents can **only generate a single action distribution** across different maps (highway and tunnel) and under various weather conditions (Default and HardRainNoon).
> On the other hand, the SRM agent adapts its behavior and **produces different action distributions** when facing various scenarios. Furthermore, the following table illustrates that, under various running seeds to mitigate random occurrences, the behavior of DrQ agents remains largely unchanged when facing drastically different environments. Hence, These two pieces of evidence verify that DrQ agents are prone to suffering from entropy collapse in this type of environment. We have added this part to the additional discussion section in Appendix F.6 (Line 875~882).
>
>
> |  Seed  | Highway(HardRain) | Highway(Hard_low_light) | Tunnel(HardRain) | Tunnel(Hard_low_light) |  std   |
> | :----: | :---------------: | :---------------------: | :--------------: | :--------------------: | :----: |
> | 1(DrQ) |      $ 14 $       |          $17$           |      $ 16 $      |         $ 17$          | $ 1.4$ |
> | 2(DrQ) |      $ 44 $       |         $ 44 $          |      $ 43 $      |         $ 43 $         | $ 0.6$ |
> | 3(DrQ) |       $ 48$       |          $ 47$          |      $ 47$       |         $ 48 $         | $ 0.6$ |
> | 1(SRM) |       $ 92$       |          $ 48$          |      $ 14$       |         $ 5 $          | $ 39 $ |
> | 2(SRM) |       $ 62$       |          $ 56$          |      $ 45$       |          $ 7$          | $ 25 $ |
> | 3(SRM) |       $ 66$       |         $ 88 $          |      $ 17 $      |          $ 1$          | $ 41 $ |

---

> ### Author Response · Authors · 2023-08-14
> **Response to Reviewer jky4 (Question 2)**
>
> **Q2:** *" DrQ and DrQ-v2 use data augmentation techniques to do representation learning. These data augmentation approaches overlap to some extent with the augmented dataset approach used in this paper. Therefore an additional discussion of this point will help the reader to understand this benchmark."*
>
> **A:**  I will clarify the differences and respective roles of these two types of augmentations. DrQ and DrQ-v2 employ data augmentation techniques: random crop and random shift, respectively. Such augmentations are referred to as **weak augmentations**, which only introduce minor changes to the image, such as slight cropping and shifting. Numerous studies [1,2,3] have shown that weak augmentations are indispensable for image-based RL to achieve high sample efficiency and good asymptotic performance. The absence of such weak augmentations could easily fail on most visual RL tasks [1,4].  However, when it comes to generalization, weak augmentations cannot help agents to obtain generalization abilities [1,5].
>
> On the other hand, the augmentation method utilizing extra datasets falls into the category of **strong augmentations** [5,6], which substantially distorts the image, such as random overlay, random conv. Contrary to weak augmentations, this approach is imperative to foster superior generalization capabilities. Nevertheless, strong augmentations will hinder the agent’s training performance. Therefore, most generalization algorithms utilize both types of augmentations [6,7,8] to achieve generalization ability while maintaining high sample efficiency.
> Thanks for your helpful suggestion. We have added this part to the additional discussion section in Appendix G (Line 883~896).
>
> ---
>
> [1] Misha Laskin, et al.Reinforcement learning with augmented data. NeurIPS, 2020.
>
> [2] Yarats D, et al. Image augmentation is all you need: Regularizing deep reinforcement learning from pixels[C]//ICLR. 2020.
>
> [3] Yarats D, et al. Mastering Visual Continuous Control: Improved Data-Augmented Reinforcement Learning[C]//ICLR. 2021.
>
> [4] Cetin E, et al. Stabilizing off-policy deep reinforcement learning from pixels[J]. arXiv preprint arXiv:2207.00986, 2022.
>
> [5] Linxi Fan, et al. Secant: Self-expert cloning for zero-shot generalization of visual policies. ICML, 2021.
>
> [6] Nicklas Hansen, et al. Stabilizing deep q-learning with convnets and vision    transformers under data augmentation. NeurIPS, 2021.
>
> [7] Zhecheng Yuan et al. Don’t touch what matters: Task-aware lipschitz data augmentation for visual reinforcement learning. arXiv preprint arXiv:2202.09982, 2022.
>
> [8] Nicklas Hansen and Xiaolong Wang. Generalization in reinforcement learning by soft data augmentation. ICRA 2021.

---

> ### Author Response · Authors · 2023-08-14
> **Response to Reviewer jky4 (Question 3)**
>
> **Q3:** *"The new benchmark as a whole falls under the category of presenting new challenges to the generalizability of reinforcement learning from the visual side. How does this new benchmark benefit the RL algorithms themselves rather than how to do representation learning better?"*
>
> **A:** First, I would like to elaborate on the intimate connection between RL algorithms and representation learning. They are coupled: An efficient RL algorithm plays an essential role in obtaining enhanced visual representations. As demonstrated in the SSL paper [1], naively applying representation learning methods for visual RL  is unlikely to have a substantial positive effect.
>
> Several visual RL algorithms [2,3,4] manage to alleviate training instability from the RL side. For example,  SVEA [4], included in our RL-ViGen baseline, optimizes the loss of the critic within the RL process, consequently delivering superior performance. Many approaches [5,6] that apply representation learning are based on SVEA to show improvement. Hence, without the development from the RL algorithm side, it is challenging to leverage the potential benefits of representation learning. Therefore, our benchmark RL-ViGen can also benefit researchers in figuring out more improved RL algorithms.
>
> Additionally, from the task aspect, our benchmark proposes a variety of challenging visuo-motor control tasks, such as quadruped robot locomotion and complex table-top manipulation, for which existing visual RL algorithms fall short. RL-ViGen can help cultivate better RL algorithms to enable agents to perform more efficiently in these tasks.
>
> ---
>
> [1] Li X, Shang J, Das S, et al. Does self-supervised learning really improve reinforcement learning from pixels?[J]. Advances in Neural Information Processing Systems, 2022, 35: 30865-30881.
>
> [2] Zhang A, McAllister R T, Calandra R, et al. Learning Invariant Representations for Reinforcement Learning without Reconstruction[C]//International Conference on Learning Representations. 2020.
>
> [3] Wu K, Wu M, Chen Z, et al. Generalizing reinforcement learning through fusing self-supervised learning into intrinsic motivation[C]//Proceedings of the AAAI Conference on Artificial Intelligence. 2022, 36(8): 8683-8690.
>
> [4] Nicklas Hansen, Hao Su, and Xiaolong Wang. Stabilizing deep q-learning with convnets and vision    transformers under data augmentation. Advances in Neural Information Processing Systems, 34, 2021.
>
> [5] Bertoin, David, et al. "Look where you look! Saliency-guided Q-networks for generalization in visual Reinforcement Learning." Advances in Neural Information Processing Systems 35 (2022): 30693-30706.
>
> [6] Yuan, Zhecheng, et al. "Pre-trained image encoder for generalizable visual reinforcement learning." Advances in Neural Information Processing Systems 35 (2022): 13022-13037.

---

> > ### Comment · Reviewer_jky4 · 2023-08-20
> >
> > Thanks to the authors for the comprehensive response.
> > My concerns have been addressed and thus I increased my score accordingly.

---

> > > ### Author Response · Authors · 2023-08-27
> > > **Response to Reviewer jky4**
> > >
> > > We are happy that our replies address your questions. We sincerely thank the reviewer for the kind response and the positive feedback.

---

### Author Response · Authors · 2023-08-14
**General Response**

We thank the reviewers for all the detailed comments and helpful suggestions. **We have highlighted the changes in blue in the revised version of our paper**. Here we provide an overview of our changes.

---

(i) Verify the claim of “DrQ can only produce a single distribution” (Appendix F.6). We plot the action distribution figure of DrQ to show that the trained agents are prone to suffer from entropy collapse.

(ii) Discuss the differences and roles between two types of data augmentation (Appendix G) .

(iii) Add a sim2real demo for responsing to Reviewer Jh47. We provide a sim2real demo to demonstrate the potential of RL-ViGen.

(iv) Structure the criteria for the difficulty level. We have elaborated on the structured criteria for the difficulty levels of the generalization types in Appendix E (Line 697 to Line 750).

(v) Illustrate environment configurations. We introduce information of environment parameters and configurations in Appendix D (Line 645 to Line 693).

(vi) Modify all sample efficiency curves. In the figures of comparing sample efficiency, we have incorporated a reference performance baseline to serve as a comparative standard.

(vii) Add an aggregated plot. We present an aggregated score for each algorithm across different types of generalization in Appendix F.5.

(viii) Add more related work. We expand our comparison with existing benchmarks, and analyze the difference between them in Appendix H.

(ix) Clarify the scope of OOD setting in abstract.



---

Website

(i) Add videos of training scenarios.

---

Repo:

Except for the helpful suggestions from reviewers, we are continuing to refine our GitHub codebase, incorporating additional elements to augment its scope and functionality.

(i) We provide an additional installation approach with Docker.

(ii) Our benchmark has been able to create a single conda environment that is suitable for all tasks.

(iii) We add extra hand embodiments to the dexterous manipulation setting.

---

### Decision · Program_Chairs · 2023-09-22

**Decision:**

Accept (Poster)

**Comment:**

This paper has received the scores 6, 7, and 8, with all reviewers unanimously voting for acceptance. They have highlighted the paper's significant contribution to advancing research in the generalization of deep reinforcement learning (RL) with high-dimensional visual inputs. Given the potential impact and the novel contributions of this study, I recommend acceptance.

Additionally, I went through the code available in the GitHub repository, and I encountered several issues that raised questions about the benchmark's reliability. It would be great if the following issues can be addressed before final submission:

1. CARLA Evaluation: The testing revealed that the weather configuration in the CARLA evaluation does not consistently function as intended. It remains uncertain whether the root of this issue lies with the CARLA simulator or the benchmark code in the repository. This inconsistency has the potential to influence the experimental results materially, necessitating a comprehensive resolution to ensure the application of accurate configurations.
2. Discrepancy in SVEA Algorithm Performance: I observed a notable discrepancy in the performance of the SVEA algorithm within the CARLA-easy environment, which registered a score of 56, markedly lower than the score cited in the paper. This deviation could potentially be influenced by the aforementioned issue and underscores the necessity to address this variance to uphold the findings' integrity.
3. Provision of Raw Scores: To record the stratified bootstrapped confidence interval for each algorithm, it is vital to provide the raw scores for each seed instead of simply offering aggregated scores in the GitHub repository.

I remain confident that, by diligently addressing these concerns, the paper can firmly establish itself as a robust contribution to the field.